# Foveal processing of emotion-informative facial features

**Nazire Duran****, Anthony P. Atkinson***

Department of Psychology, Durham University, Durham, United Kingdom

* a.p.atkinson@durham.ac.uk

## Abstract

Certain facial features provide useful information for recognition of facial expressions. In two experiments, we investigated whether foveating informative features of briefly presented expressions improves recognition accuracy and whether these features are targeted reflexively when not foveated. Angry, fearful, surprised, and sad or disgusted expressions were presented briefly at locations which would ensure foveation of specific features. Foveating the mouth of fearful, surprised and disgusted expressions improved emotion recognition compared to foveating an eye or cheek or the central brow. Foveating the brow led to equivocal results in anger recognition across the two experiments, which might be due to the different combination of emotions used. There was no consistent evidence suggesting that reflexive first saccades targeted emotion-relevant features; instead, they targeted the closest feature to initial fixation. In a third experiment, angry, fearful, surprised and disgusted expressions were presented for 5 seconds. Duration of task-related fixations in the eyes, brow, nose and mouth regions was modulated by the presented expression. Moreover, longer fixation at the mouth positively correlated with anger and disgust accuracy both when these expressions were freely viewed (Experiment 2b) and when briefly presented at the mouth (Experiment 2a). Finally, an overall preference to fixate the mouth across all expressions correlated positively with anger and disgust accuracy. These findings suggest that foveal processing of informative features is functional/contributory to emotion recognition, but they are not automatically sought out when not foveated, and that facial emotion recognition performance is related to idiosyncratic gaze behaviour.

## Introduction

### Foveal processing of emotion-informative facial features

In this study, we investigate the differential contributions of foveal and extrafoveal visual processing of facial features to the identification of facially expressed emotions [see also e.g., 1]. The fovea, a small region of the retina that corresponds to the central 1.7° of the visual field [2], is preferentially specialized for processing fine spatial detail. With increasing eccentricity from the fovea, there is a decline in both visual acuity (i.e., the spatial resolving capacity of the visual system) and contrast sensitivity (i.e., the ability to detect differences in contrast) [3,4]. Extrafoveal (or more broadly, peripheral) vision also differs qualitatively from central vision,

**Data Availability Statement:** All data files pertaining to each analysis described in the main article are available from the Open Science Framework (OSF) database via this link: https://osf.io/6n4je/.

**Funding:** The author(s) received no specific funding for this work.

**Competing interests:** The authors have declared that no competing interests exist.

receiving different processing and optimized for different tasks [4,5]. At normal interpersonal distances of approximately 0.45 to 2.1 m [6] not all features of one person's face can fall within another person's fovea at once [7]; at a distance of around 0.7 m, for example, the size of the foveated region would encompass most of one eye. Thus, at such interpersonal distances, detailed vision of another's face requires multiple fixations, bringing different features onto the fovea. Features falling outside the fovea nevertheless receive some visual processing, perhaps determining the next fixation location and even contributing directly to the extraction of socially relevant information, such as identity and emotion.

Many aspects of face perception, including the recognition of emotional expressions, rely on a combination of part- or feature-based and configural or holistic processing [8,9]. Although configural or holistic processing appears to contribute more to face perception at normal interpersonal distances than at larger distances [10–12, though see 13], the extraction of information from individual facial features nevertheless helps underpin a range of face perception abilities, including facial emotion recognition, even when the face subtends a visual angle equivalent to a real face viewed at a normal interpersonal distance. Findings from studies that involved presenting observers with face images filtered with randomly located Gaussian apertures or "Bubbles" whose size and corresponding spatial frequency content varied have shown that certain facial features are more informative than others for recognition and discrimination of each basic emotional expression from the others [14–17]. For anger, for example, the brow region is informative, especially at mid-to-high spatial frequencies (15–60 cycles per image, where the face takes up most of the image); for fear, the eye region is informative, especially at high spatial frequencies (60–120 cycles per image), and the mouth at lower spatial frequencies (3.8–15 cycles per image); for disgust, the mouth and sides of the nose are most informative, especially at the middle to the highest spatial frequencies (15–120 cycles per image). The relative contribution to emotion classification performance of these emotion-informative regions and of the spatial frequency information at those regions can vary depending on the combination of emotions used in the task, however [16,17].

In a previous study, we showed that emotion classification performance varies according to which emotion-informative facial feature is fixated [1]. In that study, we used a slightly modified version of a 'brief-fixation paradigm' developed by Gamer and Büchel [18], in which faces were presented for 80 ms, a time insufficient for a saccade, at a spatial position that guaranteed that a given feature–the left or right eye, the left or right cheek, the central brow, or mouth–fell at the fovea. In one experiment, participants classified angry, fearful, happy and emotionally neutral faces, and in another experiment a different group of participants classified angry, fearful, surprised and neutral faces. Across both experiments, observers were more accurate and faster at discriminating angry expressions when the brow was projected to their fovea than when one or other cheek or eye (but not mouth) was, an effect that was principally associated with a reduction in the misclassifications of anger as emotionally neutral. This finding is consistent with the importance of mid-to-high spatial-frequency information from the brow in allowing observers to distinguish angry expressions from expressions of other basic emotions [14–17]. Yet, in the first experiment, performance in classifying fear and happiness was not influenced by whether the most informative features (eyes and mouth, respectively) were projected foveally or extrafoveally. In the second experiment, observers more accurately classified fearful and surprised expressions when the mouth was projected to the fovea, effects that were principally associated with reductions in the number of confusions between these two emotions. This enhanced ability to distinguish between fearful and surprised expressions when the mouth was fixated is consistent with previous work showing that the mouth distinguishes fearful from surprised as well as from neutral and angry expressions, whereas the eyes and brow do not distinguish between prototypical fearful and surprised expressions [14,15,19,20].

In the context of the brief-fixation paradigm, while the effect of enforced fixation on a facial feature informs us about whether foveal processing of that feature is beneficial for recognition of target expressions, the subsequent saccades might inform us about what extrafoveal information is being sought out following initial fixation on a facial feature. Previous research using the brief-fixation paradigm revealed that there was a higher proportion of reflexive first saccades upwards from fixation on the mouth than downwards from fixation on the eyes or on a midpoint between the eyes and that this effect was modified by the viewed emotion [18,21–26]. In these experiments, observers classified either angry, fearful, happy and neutral faces, or just fearful, happy and neutral faces, which were presented for 150 ms. Reflexive first saccades were defined as the first saccades from enforced fixation on the face that occurred within 1 s of face offset. The greater propensity for observers to saccade upwards from fixation on the mouth than downwards from fixation on or between the eyes tended to be evident for fearful, neutral and angry faces and markedly reduced or absent for happy faces. Might these saccades, which were presumably triggered in response to the face, be targeting the informative facial features of the expression (e.g., the eyes of the fearful expressions) when not initially fixated? That is the implication in at least some of these previous studies, in which the authors summarize the upward saccades from the mouth as 'toward the eyes' and the downward saccades from the eyes as 'toward the mouth', and even sometimes claim that their findings show that people reflexively saccade toward diagnostic emotional facial features [see especially 18,22,26]. The suggestion that the appearance of emotion-informative facial features outside foveal vision can trigger reflexive eye-movements towards these features gains some support from a study by Bodenschatz et al. [27]: when primed with fearful faces, observers fixated the eye region of a subsequently presented neutral face quicker and dwelled on this region for longer whereas they fixated the mouth region quicker and dwelled on this region for longer when primed with a happy face compared to a sad prime.

Yet, in the context of the brief-fixation paradigm, our previous study [1] did not find any evidence to indicate that reflexive first saccades preferentially target emotion-distinguishing facial features. Although we replicated the key finding of more upward saccades from enforced fixation on the mouth than downward saccades from enforced fixation on the eyes, the modulation of this effect by the expressed emotion was evident in our first experiment, in which angry, fearful, happy and neutral faces were used (the effect was evident for angry and neutral faces, less so for fearful faces, and not at all for happy faces), but not in our second experiment, in which angry, fearful, surprised and neutral faces were used. Moreover, in an attempt to provide a more spatially precise measure of the extent to which reflexive first saccades target diagnostic features, we calculated a saccade path measure by projecting the vector of each first saccade on to the vectors from the enforced fixation location to each of the other facial locations of interest, normalized for the length of the target vector (given that the target locations vary in distance from a given fixation location). Using this measure, we did not find any support for the hypothesis that observers' first saccades from initial fixation on the face will seek out emotion-distinguishing features. Instead, we found that reflexive first saccades tended towards the left and centre of the face, which might reflect one or more of (a) a centre-of-gravity effect, that is, a strong tendency for first saccades to be to the geometric center of scenes or configurations [e.g., 28–31], including faces [e.g., 32], (b) the left visual field/right hemisphere advantage in emotion perception [e.g., 33–35] and (c) the strong tendency for first saccades onto a face to target a location below the eyes, just to the left of face centre, which is also the optimal initial fixation point for determining a face's emotional expression [36].

In the first two of three experiments reported in the present paper, we extend our previous work by using the same brief-fixation method to investigate whether fixating an informative facial feature improves emotion recognition over fixating non-informative facial features, here

using different combinations of emotion: angry, fearful, surprised and sad faces in Experiment 1 and angry, fearful, surprised and disgusted faces in Experiment 2a. We aimed particularly to answer the following questions: (1) Would fixation on the brow of angry faces improve emotion classification accuracy even when 'neutral' was no longer a response option or stimulus condition? (2) Would the enhanced ability to distinguish between fearful and surprised expressions when fixating the mouth still be evident when those expressions are presented along with different combinations of emotions? (3) Would fixation on the brow or mouth enhance classification accuracy for sad expressions, relative to fixation on an eye or cheek, given that observers rely on medium-to-high spatial frequency information at both these regions to classify sad faces [14–17]? (4) Would fixation on the mouth enhance classification accuracy for disgusted expressions, relative to fixation on the brow, an eye or cheek, given that observers rely on medium-to-high spatial frequency information on and around the mouth to classify disgusted faces [14–17]? We also here extend our previous work [1] by examining the direction and paths of reflexive first saccades triggered by the onset of the briefly presented face, to further test the hypothesis that those saccades target emotion-informative facial features. Although we found no support for this hypothesis in our previous study, additional tests of this hypothesis with different combinations of emotions are warranted.

In the last of the three experiments reported in this paper (Experiment 2b), we presented faces for a longer duration, thus allowing observers to freely fixate multiple locations on the face. Our aims were to investigate whether different facial expressions elicit specific fixation patterns under task instructions to classify the expression and whether these fixation patterns are related to emotion classification performance. A small number of eye-tracking studies indicate that emotion-informative facial features modulate eye movements when observers view images of facial expressions [26,37–40]. However, these studies either did not require participants to classify the facially expressed emotion or, when they did, the relationship between the amount of time spent fixating informative features and emotion recognition performance was not examined. These studies therefore do not address whether the increased time spent fixating an informative facial feature contributes to the accuracy of facial expression recognition, except for one recent study [41], which found that when the observers misclassified expressions, they explored regions of the face that supported the recognition of the mistaken expression.

Nonetheless, there is some, albeit limited, evidence that fixation of emotion-distinguishing features whilst free-viewing faces aids emotion recognition. Notably, a selective impairment in recognizing fear from faces associated with bilateral amygdala damage is the result of a failure to saccade spontaneously to and thus fixate the eye region [42], a region that is informative for fear [15,17]. Remarkably, instructing the patient with bilateral amygdala damage to fixate the eyes restored fear recognition performance to normal levels [42]. Recognition of negatively-valenced emotions (i.e., fear, anger, sadness), especially fear, was shown to be impaired in individuals with Autism Spectrum Disorders (ASDs) and the impairment is linked to a decreased preference to fixate or saccade to the eye region of the facial expressions [25,43]. Another study with neurotypical participants found that accuracy in detecting emotional expressions was predicted by participants' fixation patterns, particularly with fixations to the eyes and, to a lesser extent, the nose and mouth, though mostly for subtle rather than strong expressions [44]. These studies suggest that fixating on the informative facial features aids successful emotion recognition and failure to do so leads to impairments in emotion recognition. Yet it remains unclear the extent to which this reflects an emotion-specific pattern (i.e., more or longer fixations for one feature for a particular emotion and for a different feature for a different emotion) as opposed to a pattern reflecting the relative importance of particular facial features irrespective of the emotion. Moreover, to our knowledge, existing evidence does not support

an alternative hypothesis, namely, that because observers are more efficient at extracting task-relevant information from emotion-distinguishing facial features they fixate those features less.

## Experiment 1

In this experiment, we used the same modified version of Gamer and Büchel's [18] brief-fixation paradigm used by Atkinson and Smithson [1], here with angry, fearful surprised and sad expressions, to test two hypotheses: (1) A single fixation on an emotion-distinguishing facial feature will enhance emotion identification performance compared to when another part of the face is fixated (and thus the emotion-informative feature is projected to extrafoveal retina). (2) Under these task conditions, observers' reflexive first saccades from initial fixation on the face will preferentially target emotion-distinguishing facial features when those features are not already fixated.

Based on previous work showing that the brow is informative for anger recognition [14–17] and the mouth for distinguishing between fear and surprise [14,15,20], and in line with our previous findings [1], we predicted that enforced fixation on (1) the central brow would improve emotion recognition for anger compared to other initial fixation locations, and (2) the mouth would improve emotion recognition accuracy for fearful and surprised expressions. Based on previous research showing that the brow and mouth are informative for the recognition of sadness [14–16], we also predicted that (3) enforced fixation on the brow and mouth would improve accuracy for sad expressions, relative to fixation on a cheek and either eye. The cheeks were chosen as fixation locations to be relatively uninformative regions for the chosen expressions.

To test the second hypothesis, that reflexive first saccades would target emotion-informative facial features, we used a measure of saccade direction that mapped the direction of the reflexive saccades onto six possible saccade paths leading to the target facial features. If the reflexive saccades do indeed target emotion-informative facial features, then the paths of those saccades would be expected to be more similar to the paths leading to the relevant informative features than to the paths leading to the less-informative features. On the other hand, the direction of the reflexive saccades might instead reflect a central or a leftward bias, or both, as we have found previously [1].

## Method

**Participants.**   Thirty-three participants took part (31 female; mean age = 20.7 years, age range = 18–31). All participants were undergraduate Psychology students and had normal or corrected-to-normal vision. All participants gave written consent to take part in the experiment and were awarded participant pool credit for their participation. The study was approved by the Durham University Psychology Department Ethics Sub-committee.

**Materials.**   Twenty-four facial identities (12 males, 12 females) were chosen from the Radboud Faces Database [45]. All faces were of White adults with full frontal pose and gaze. Angry, fearful, surprised and sad expressions from each identity were utilised leading to a total of 96 images. The images used were spatially aligned, as detailed in [45]. The images were presented in colour and were cropped from their original size to 384 (width) × 576 (height) pixels, so that the face took up more of the image than in the original image set. Each image therefore subtended 14.9 (w) × 22.3 (h) degrees of visual angle at a viewing distance of 57cm on a 1024 × 768 resolution screen.

**Design.**   A within-subjects design was used, with Expression (anger, fear, surprise and sadness) and Fixation Location (eyes, brow, cheeks, and mouth) as repeated-measures variables.

There were 4 blocks of 96 trials. Over the course of 4 blocks, participants were presented with each emotion at each of the 4 initial fixation locations 24 times (once for each identity), with the eye and the cheek fixation locations selected equally (12 times) on the left and right. The stimuli were randomly ordered across the 4 blocks, with a new random order for each participant.

**Apparatus and procedure.** The experiment was executed and controlled using the Matlab® programming language with the Psychophysics Toolbox extensions (Brainard, 1997; Pelli, 1997; Kleiner et al, 2007). To control stimulus presentation and to measure gaze behaviour, we used an EyeLink 1000 desktop-mounted eye-tracker (SR Research, Mississauga, ON, Canada). Each stimulus block was started with the default nine-point calibration and validation sequences. Recording was binocular but only data from the left eye was analysed since data from the left eye started the trial in a gaze-contingent manner. Default criteria for fixations, blinks, and saccades implemented in the Eyelink system were used.

Each trial started with a fixation cross located at one of 25 possible locations on the screen. This was to make both the exact screen location of the fixation cross and the to-be-fixated facial feature unpredictable. These 25 possible locations for the fixation cross were at 0, 25, or 50 pixels left or right and up or down from the center of the screen. These fixation-cross positions were randomly ordered across trials. Following the presentation of the fixation cross on one of these randomly selected screen locations, a face image was presented so that one of the facial features of interest was aligned with the location of that fixation cross. Faces were presented in a gaze-contingent manner on each trial: The participants needed to fixate within 30 pixels (1.16 degrees of visual angle) of the fixation cross for 6 consecutive eye-tracking samples following which a face showing one of the four target expressions was presented. The facial features of interest (which henceforth we refer to as 'fixation locations') were the left or right eye, the central brow, the centre of the mouth, and locations on the left and right cheeks. These initial fixation locations can be seen in Fig 1. The face was presented for 82.4 ms (7 monitor refreshes) on a monitor with an 85 Hz refresh rate. Following the face presentation, the participant pressed a key on a QWERTY keyboard to indicate their answer. The row of number keys near the top of the keyboard were used, with 4 for anger, 5 for fear, 8 for surprise and 9. The keys were labelled A, F and Su and Sa from left to right and the order of these keys remained the same for each participant. Participants pressed the A and F keys with the left and the Su and Sa keys with the right hand. This configuration was chosen to optimize the reach of the participants to the keyboard from either side of the chinrest. Participants were asked to memorize the keys so as not to look down towards the keyboard during the experiment. A valid response needed to be registered for the next trial to begin.

A                                                    B

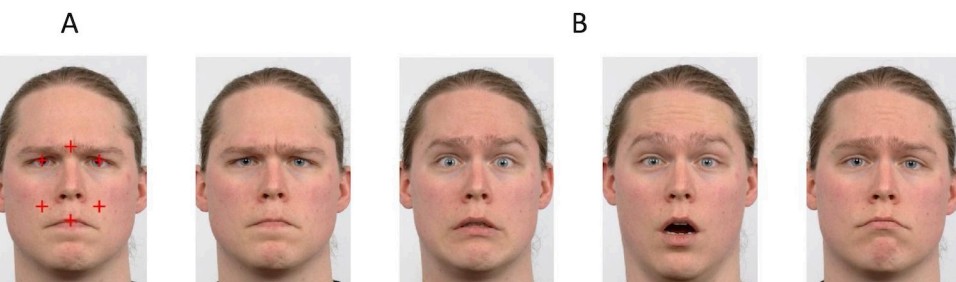

**Fig 1. Initial fixation locations and example facial expressions used.** (A) An example face image used in the experiments (from the [45] database), overlaid, for illustrative purposes, with red (dark grey) crosses to mark the possible enforced fixation locations. (B) Example images of each expression used in Experiment 1 (left to right: anger, fear, surprise, sadness). The face images are republished in slightly adapted form from the Radboud Faces Database [45] under a CC BY license, with permission from Dr Gijsbert Bijlstra, Radboud University.

## Data analysis

Our main analyses consisted in repeated measures ANOVAs and planned comparison t-tests. Where the assumption of sphericity was violated, the Greenhouse-Geisser corrected F-value is reported. Significant interactions were followed up by simple main effects analyses (repeated measures). Significant main effects were followed up with pairwise comparisons. Planned comparisons (one-tailed paired samples t-tests) were conducted to test the hypotheses relating to the fixation location specific to each expression, which led to three comparisons for each expression, i.e., for anger: brow > eyes, brow > mouth, brow > cheeks; and for fear, surprise and disgust: mouth > eyes, mouth > brow, mouth > cheeks. All planned and pairwise comparisons were corrected for multiple comparisons using the Bonferroni-Holm method (uncorrected p-values are reported). For ANOVAs, we report both the partial eta squared ($\eta_p^2$) and generalized eta squared ($\eta_G^2$) measures of effect size [46–48]; for the partial eta squared values we also report 90% confidence intervals (CIs) calculated using code extracted from the 'ci.pvaf' function in the MBESS package (ver. 4.6.0) for R [49,50]. For t-tests, we report Cohen's $d_z$ effect size with 95%, as output by JASP. The data are summarised in rain-cloud plots [51] using Matlab code from *https://github.com/RainCloudPlots/*, supplemented with error bars indicating the 95% CIs calculated in JASP (ver 0.10.2) using bias-corrected accelerated boot-strapping with 1000 samples.

*Emotion classification accuracy*. Trials with reaction times shorter than 200 ms were disregarded as automatic responses not reflecting a genuine perceptual response. For this experiment, there were no such outliers. Unbiased hit rates were calculated using the formula supplied by Wagner [52]. The "unbiased hit rate" (Hu) accounts for response biases in classification experiments with multiple response options [52]. Hu for each participant is calculated as the squared frequency of correct responses for a target emotion in a particular condition divided by the product of the number of stimuli in that condition representing this emotion and the overall frequency that that emotion category is chosen for that condition. Hu ranges from 0 to 1, with 1 indicating that all stimuli in a given condition representing a particular emotion have been correctly identified and that that emotion label has never been falsely selected for a different emotion.

Prior to analysis, the unbiased hit rates were arc sine square-root transformed in order to better approximate a normal distribution of the accuracy data. Emotion recognition accuracy was analysed by a $4 \times 4$ repeated measures ANOVA with emotion and initial fixation location as factors. Unless otherwise stated, the data from the left and right sides of the face (when the fixation location was an eye or cheek) were collapsed for the analyses.

*Eye-movement analysis*: *Reflexive saccade selection criteria*. Reflexive saccades were defined as saccades that were triggered by onset of a face image and executed following face offset. Accordingly, reflexive saccades used for the reported analyses were chosen as those that happened within the 82.4 ms to 1000 ms window after face onset [similar to e.g., 18,26]. In other words, the reflexive saccades used were the ones that were executed within 1000ms after face offset. All reflexive saccades that had amplitudes smaller than 0.5 degrees were disregarded. Finally, of all the saccades that complied with these criteria, only the first saccade was used. After this data reduction, participants who had reflexive first saccades on fewer than 20% of the total trials per block (i.e., 20% of 96) in any one of the 4 blocks were removed from further analysis. This threshold was chosen to strike a balance between having enough trials per condition and not wanting to exclude the data for too many participants. All saccade direction related analyses were carried out on this set of data.

*Eye-movement analysis*: *Proportions up vs. down*. These analyses are reported in the Supplementary Results and Discussion in S1 File, to allow comparison with previous studies [18,21–26].

*Eye-movement analysis*: *Saccade paths*. To estimate the paths of the reflexive saccades, six vectors were plotted from the starting coordinates of each saccade to the coordinates of the six possible initial fixation locations. These make up the saccade path vectors. Then, the dot products of the reflexive saccade vector and the six possible saccade path vectors were calculated and normalised to the magnitude of the saccade path vectors. This measure represents the similarity between the reflexive saccade path and the possible saccade path vectors. Identical vectors would produce a value of 1, whereas a saccade in exactly the same direction as a possible saccade path to a target but overshooting that target by 50% would produce a value of 1.5. Negative saccade path values, on the other hand, indicate a saccade that is going in the opposite direction of the possible saccade path. Trials with > 3 out of the 5 normalized saccade paths (excluding trajectories from the initial fixation location to itself) whose absolute values > 1.5 were classed as outliers. This effectively excluded saccades that ended beyond the edge of the face. Such outliers accounted for 0.85% of the recorded measures and were excluded from the analyses.

## Results

**Emotion classification accuracy.** The unbiased hit rates are summarized in Fig 2. There was a main effect of emotion, $F(1.66, 53.06) = 17.02$, $p < .001$, $\eta p^2 = .35$, 90% CI [.17 .47], $\eta_G^2 = .1$, reflecting worse accuracy for fear compared to the other emotions (all uncorrected $p$s $< .001$). A main effect of fixation location, $F(3, 96) = 3.43$, $p = .02$, $\eta p^2 = .1$, 90% CI [.01 .18], $\eta_G^2 = .02$, reflected higher recognition accuracy for fixation on the mouth ($M = 0.722$, $SD = 0.114$) compared to the cheeks ($M = 0.685$, $SD = 0.116$), $p = .006$, $d_z = 0.51$ (Bonferroni-Holm adjusted $\alpha = .0083$) and eyes ($M = 0.685$, $SD = 0.116$), $p = .009$, $d_z = 0.48$ (Bonferroni-Holm adjusted $\alpha = .01$). Importantly, there was a significant interaction between these factors, $F(5.49, 175.63) = 3.68$, $p = .003$, $\eta p^2 = .1$, 90% CI [.02 .15], $\eta_G^2 = .02$, which we followed up with simple main effects analyses.

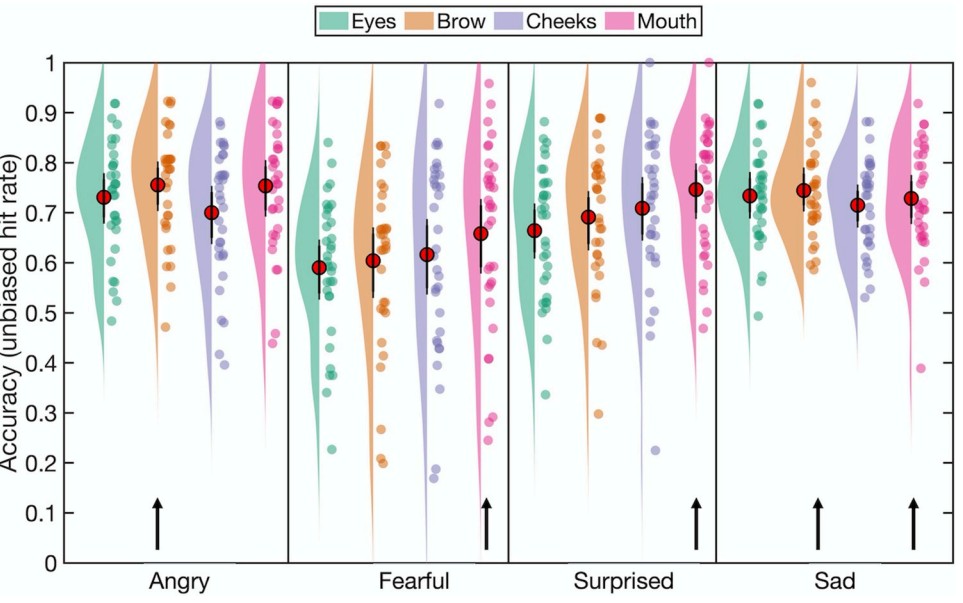

**Fig 2. Emotion classification accuracy (mean unbiased hit rates) as a function of emotion category and fixation location in Experiment 1.** Red circles indicate the mean value across participants and error bars indicate the 95% CIs (see Methods). The raincloud plot combines an illustration of data distribution (the 'cloud') with jittered individual participant means (the 'rain') for each condition [51].

There was a main effect of fixation location for angry expressions, $F(3,96) = 3.14$, $p = .029$, $\eta p^2 = .09$, 90% CI [.005 .17], $\eta_G^2 = .04$. Planned comparisons revealed that anger recognition accuracy with fixation at the brow ($M = 0.755$, $SD = 0.112$) was higher compared to fixation at the cheeks ($M = 0.7$, $SD = 0.132$), $t(32) = 2.45$, $p = .01$, $d_z = 0.43$, 95% CI [0.12 $\infty$] (Bonferroni-Holm adjusted $\alpha = .0167$). The full pairwise comparisons additionally revealed that fixation on the mouth ($M = 0.754$, $SD = 0.127$) led to significantly higher anger accuracy compared to fixation on the cheeks, $t(32) = 2.92$, $p = .006$, $d_z = 0.51$, 95% CI [0.14 0.87] (Bonferroni-Holm adjusted $\alpha = .0083$).

There was a main effect of fixation location for fearful faces, $F(3, 96) = 3.83$, $p = .012$, $\eta p^2 = .11$, 90% CI [.01 .19], $\eta_G^2 = .03$. Planned comparisons showed that fixation on the mouth ($M = 0.658$, $SD = 0.185$) led to improved fear recognition compared both to the eyes ($M = 0.59$, $SD = 0.141$), $t(32) = 2.98$, $p = .003$, $d_z = 0.52$, 95% CI [0.21 $\infty$] and the brow ($M = 0.604$, $SD = 0.171$), $t(32) = 2.55$, $p = .008$, $d_z = 0.44$, 95% CI [0.14 $\infty$]. Fear recognition accuracy was also greater with fixation on the mouth than on the cheeks ($M = 0.616$, $SD = 0.192$), though this effect did not survive correction for multiple comparisons, $t(32) = 2.04$, $p = .025$, $d_z = 0.36$, 95% CI [0.06 $\infty$]. The full pairwise comparisons did not reveal any additional differences between the fixation locations.

There was also a main effect of fixation location for surprised faces, $F(3, 96) = 5.78$, $p = .001$, $\eta p^2 = .15$, 90% CI [.04 .24], $\eta_G^2 = .05$. Planned comparisons confirmed higher accuracy for surprise with fixation at the mouth ($M = 0.746$, $SD = 0.129$) compared to an eye ($M = 0.664$, $SD = 0.132$), $t(32) = 4.28$, $p < .001$, $d_z = 0.75$, 95% CI [0.42 $\infty$] and the brow ($M = 0.691$, $SD = 0.14$), $t(32) = 3.19$, $p = .002$, $d_z = 0.56$, 95% CI [0.24 $\infty$]. There was no difference in accuracy between fixation at the mouth and fixation at a cheek ($M = 0.709$, $SD = 0.154$), $t(32) = 1.51$, $p = .071$. The full pairwise comparisons did not reveal any additional differences between the fixation locations. Finally, there was no effect of initial fixation location for sad expressions ($F < 1$, $p > .4$).

To further investigate the effect of fixation location, we also examined the confusion matrices to identify whether there were systematic misclassifications of the target expressions as another emotion (see Table 1). Anger was most often misclassified as sadness, but this misclassification was reduced when angry expressions were presented with initial fixation at the brow. Fearful expressions tended to be misclassified as surprised and this misclassification was reduced when the fearful expression was presented with initial fixation at the mouth.

**Table 1. Confusion matrices for Experiment 1 (brief fixation paradigm).**

|  | Anger | Fear | Surprise | Sadness | Anger | Fear | Surprise | Sadness |
|---|---|---|---|---|---|---|---|---|
|  | Eyes |  |  |  | Brow |  |  |  |
| Anger | 84.22 | 2.90 | 1.64 | **11.24** | 86.74 | 2.15 | 1.77 | **9.34** |
| Fear | 3.66 | 70.45 | **22.98** | 2.90 | 3.41 | 69.07 | **23.23** | 4.29 |
| Surprise | 1.14 | 10.48 | 86.24 | 2.15 | 1.26 | 7.95 | 88.89 | 1.89 |
| Sadness | 9.22 | 1.64 | 2.90 | 86.24 | 9.34 | 1.39 | 2.53 | 86.74 |
|  | Cheeks |  |  |  | Mouth |  |  |  |
| Anger | 82.32 | 2.53 | 1.01 | **14.14** | 85.98 | 1.52 | 0.25 | **12.25** |
| Fear | 4.17 | 71.46 | **20.58** | 3.79 | 2.78 | 74.87 | **16.79** | 5.56 |
| Surprise | 1.64 | 9.22 | 87.88 | 1.26 | 1.39 | 8.59 | 89.14 | 0.88 |
| Sadness | 10.48 | 1.64 | 1.64 | 86.24 | 9.47 | 1.39 | 1.89 | 87.25 |

One confusion matrix is shown for each of the 4 initial fixation locations (eyes, brow, cheeks, mouth). The row labels indicate the presented expression and the column labels indicate the participant responses. The data are the % of trials each emotion category was given as the response to the presented expression. %s reported in bold represent the most prevalent confusions for each expression.

**Eye movement analysis.** We conducted analyses of saccade direction using the saccade path measure described in Data Analysis to compare the paths of reflexive saccades to possible saccade paths targeting one of the other initial fixation locations. We first analysed the saccade path measures collapsed across fixation location and then for each fixation location separately. If reflexive first saccades target emotion-distinguishing features, then we would expect those saccades to be directed more strongly towards the brow and possibly also the mouth for angry and sad faces, and more strongly towards the mouth for fearful and surprised faces.

Reflexive saccade data from three participants were removed from the analysis of eye-movement data since they did not meet the criteria for inclusion. The following analyses are conducted on 30 participants. On average, for the 30 participants, 70% (range: 32%-98%) of all trials included a reflexive saccade.

*Saccade path analysis*: *Collapsed across fixation location*. The mean saccade paths of the first saccades were calculated for each emotion and collapsed across all the initial fixation locations. A visualisation of these data can be seen in Fig 3A. A 4 × 6 repeated measures ANOVA was

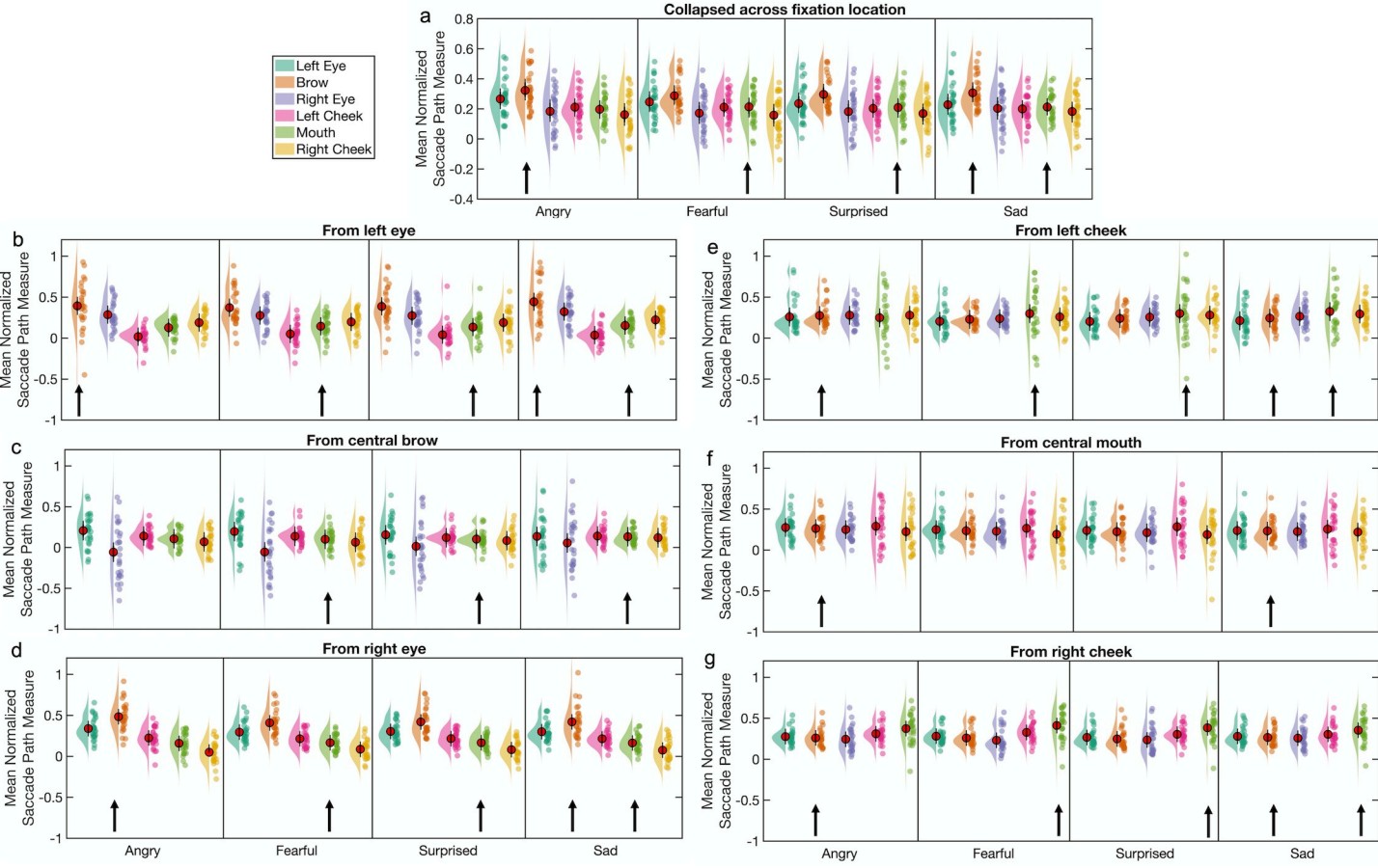

**Fig 3. Mean normalized saccade paths as a function of facial expression and target location for Experiment 1.** The normalized saccade path is a measure of the directional strength of the reflexive first saccades (executed after face offset) towards target locations of interest, in this case (a) to 6 target locations, collapsed across initial fixation location (N = 30), and from (b) the left eye, (c) the brow, (d) the right eye, (e) the left cheek, (f) the mouth, and (g) the right cheek, to the remaining 5 regions of interest (N = 27). Red circles indicate the mean value across participants and error bars indicate the 95% CIs (see Methods). The raincloud plot combines an illustration of data distribution (the 'cloud') with jittered individual participant means (the 'rain') for each condition [51]. Arrows indicate the most emotion-informative ('diagnostic') facial features for each emotion.

used to compare the mean saccade paths towards each of the six possible saccade targets for each expression. A main effect of target location, $F(1.82, 52.82) = 8.71$, $p < .001$, $\eta p^2 = .23$, 90% CI [.07 .36], $\eta_G^2 = .13$, was modified by an interaction with emotion, $F(7.39, 214.44) = 4.76$, $p < .001$, $\eta p^2 = .14$, 90% CI [.05 .19], $\eta_G^2 = .007$. There was a negligible main effect of emotion ($F = 1.58$, $p = .2$). Simple main effects analyses revealed differences in the mean normalized saccade path values between target locations for all 4 emotions (all $Fs > 6.5$, $ps < .001$) and between emotions for the 3 upper-face target locations (left eye, brow, right eye; $Fs > 3.9$, $ps < .015$) but not for the 3 lower-face target locations (left cheek, mouth, right cheek; $Fs < 2.5$, $ps > .05$). Pairwise comparisons to follow up the main effect of target location revealed that reflexive first saccades were more strongly in the direction of the brow ($M = 0.303$, $SD = 0.111$) than of any of the other 5 target locations: left eye ($M = 0.244$, $SD = 0.117$), $t(29) = 3.87$, $p < .001$, $d_z = 0.71$, 95% CI [.3 1.1]; right eye ($M = 0.185$, $SD = 0.134$), $t(29) = 6.17$, $p < .001$, $d_z = 1.13$, 95% CI [.66 1.58]; left cheek ($M = 0.206$, $SD = 0.098$), $t(29) = 5.22$, $p < .001$, $d_z = 0.95$, 95% CI [.51 1.38]; mouth ($M = 0.208$, $SD = 0.101$), $t(29) = 4.13$, $p < .001$, $d_z = 0.76$, 95% CI [.34 1.16]; right cheek ($M = 0.167$, $SD = 0.124$), $t(29) = 6.12$, $p < .001$, $d_z = 1.12$, 95% CI [.65 1.57]. This finding that saccades were more strongly in the direction of the brow than of any of the other 5 target locations was evident for all 4 emotions, as revealed by examining pairwise comparisons for each emotion separately, though after correction for multiple comparisons, this was not the case for the brow compared to the mouth and left eye of fearful faces (respectively, $t = 2.94$, $p = .006$, adjusted $\alpha = .0046$, and $t = 2.8$, $p = .009$, adjusted $\alpha = .0056$; all other $ts > 3.5$, $ps \leq .001$). Collapsed across emotion, reflexive saccades were also more strongly in the direction of the left eye than of the left cheek, $t(29) = 3.72$, $p < .001$, $d_z = 0.68$, 95% CI [.28 1.07], an effect that was also evident for each emotion when examined separately, but only for angry and surprised faces after correction for multiple comparisons (respectively, $t = 4.68$, $p < .001$, adjusted $\alpha = .0046$, and $t = 3.15$, $p = .004$, adjusted $\alpha = .005$; all other $ts < 3.2$, $ps > .003$).

Another set of simple main effects analyses, comparing the mean normalized saccade path values between emotions for the brow and mouth target locations separately, revealed an effect of emotion for the brow target location only, $F(2.33, 67.49) = 7.08$, $p < .001$, $\eta p^2 = .2$, 90% CI [.06 .31], $\eta_G^2 = .01$ (other $F = 1.17$, $p = .32$). Pairwise comparisons revealed that reflexive first saccades were more strongly directed towards the brow for angry faces, $t(29) = 4.27$, $p < .001$, $d_z = 0.78$, 95% CI [.36 1.18], and sad faces, $t(29) = 2.82$, $p = .009$, $d_z = 0.51$, 95% CI [.13 .89], than for fearful faces.

*Saccade path analysis*: *From fixation on individual features*. Next, we conducted separate $4 \times 5$ repeated measures ANOVAs for each of the 6 fixation locations, with emotion and saccade target as factors. The data are summarized in Fig 3B–3F. Note that the data for a further 3 participants had to be excluded from these analyses, due to missing data in one or more cells of the design matrices; thus N = 27 for these analyses.

For saccades starting from fixation at the left eye, there was a significant effect of target location, $F(1.07, 27.84) = 54.47$, $p < .001$, $\eta p^2 = .68$, 90% CI [.48 .77], $\eta_G^2 = .35$. All pairwise comparisons were significant (see S1 Table). Reflexive first saccades were most strongly directed towards the brow ($M = 0.399$, $SD = 0.233$), followed by the right eye ($M = 0.289$, $SD = 0.162$), right cheek ($M = 0.199$, $SD = 0.127$), mouth ($M = 0.14$, $SD = 0.112$) and left cheek ($M = 0.037$, $SD = 0.099$). This effect was not modified by an interaction with emotion ($F = 1.21$, $p = .3$), as would be expected if reflexive first saccades targeted emotion-distinguishing features. There was a negligible effect of emotion ($F = 1.64$, $p = .21$).

For saccades starting from fixation at the right eye, there was a significant effect of target location, $F(1.15, 29.79) = 108.78$, $p < .001$, $\eta p^2 = .81$, 90% CI [.68 .86], $\eta_G^2 = .48$, but no effect of emotion ($F < 1$, $p > .4$). All pairwise comparisons for the effect of target location were significant (see S1 Table). Reflexive first saccades were most strongly directed towards the brow

                                                    

($M = 0.433$, $SD = 0.162$), followed by the left eye ($M = 0.309$, $SD = 0.117$), left cheek ($M = 0.217$, $SD = 0.098$), mouth ($M = 0.163$, $SD = 0.094$) and right cheek ($M = 0.073$, $SD = 0.106$). In contrast to the results for saccades from the left eye, the effect of target location for saccades from the right eye was modified by an interaction with emotion, $F(3.48, 90.56) = 3.75$, $p = .01$, $\eta p^2 = .13$, 90% CI [.02 .21], $\eta_G^2 = .014$. Simple main effects analyses showed that the effect of target location was significant for all 4 emotions ($Fs > 48.0$, $ps < .001$). Pairwise comparisons showed the same pattern for all 4 emotions as for the main effect (brow > left eye > left cheek > mouth > right cheek). Interestingly, pairwise comparisons across emotions for the brow target location revealed that reflexive first saccades from fixation at the right eye were directed more strongly towards the brow for angry faces ($M = 0.483$, $SD = 0.178$) than for fearful ($M = 0.408$, $SD = 0.157$), $t(26) = 3.98$, $p < .001$, $d_z = 0.77$, 95% CI [0.33 1.19], surprised ($M = 0.419$, $SD = 0.166$), $t(26) = 3.12$, $p = .004$, $d_z = 0.6$, 95% CI [0.19 1.01], and sad faces ($M = 0.42$, $SD = 0.195$), $t(26) = 2.87$, $p = .008$, $d_z = 0.55$, 95% CI [0.14 0.95].

For saccades starting from fixation at the brow, there was a marginal effect of target location, $F(1.05, 27.16) = 4.09$, $p = .052$, $\eta p^2 = .14$, 90% CI [NA .33], $\eta_G^2 = .09$, and a negligible effect of emotion ($F = 1.8$, $p = .15$). Numerically, reflexive first saccades were more strongly directed towards the left eye ($M = 0.173$, $SD = 0.204$) and left cheek ($M = 0.135$, $SD = 0.094$) than towards the mouth ($M = 0.11$, $SD = 0.096$), right cheek ($M = 0.084$, $SD = 0.129$) and right eye ($M = -0.01$, $SD = 0.306$), but none of the pairwise comparisons was significant after correction for multiple comparisons (all uncorrected $ps \geq .02$, $d_z s \leq 0.47$). Nonetheless, the effect of target location for saccades from the brow was modified by an interaction with emotion, $F(3.3, 85.88) = 5.25$, $p = .002$, $\eta p^2 = .17$, 90% CI [.04 .26], $\eta_G^2 = .017$. Simple main effects analyses showed an effect of target location for angry faces, $F(4) = 6.96$, $p < .001$, and fearful faces, $F(4) = 6.68$, $p < .001$, a marginal effect for surprised faces, $F(4) = 2.39$, $p = .056$, and no effect for sad faces, $F(4) = 0.83$, $p = .5$. Reflexive first saccades from the brow of angry faces were more strongly in the direction of the mouth ($M = 0.107$, $SD = 0.103$) than of the right cheek ($M = 0.069$, $SD = 0.138$), $t(26) = 3.13$, $p = .004$, $d_z = 0.6$, 95% CI [0.19 1.01] (Bonferroni-Holm adjusted $\alpha = .005$), and marginally more strongly towards the left cheek ($M = 0.141$, $SD = 0.098$) than towards the right cheek, $t(26) = 3.02$, $p = .006$, $d_z = 0.58$, 95% CI [0.17 0.99] (Bonferroni-Holm adjusted $\alpha = .0056$) and the mouth, $t(26) = 2.9$, $p = .007$, $d_z = 0.58$, 95% CI [0.15 0.96] (Bonferroni-Holm adjusted $\alpha = .0063$). Reflexive first saccades from the brow of fearful faces were more strongly in the direction of the three lower-face features than of the right eye ($M = -0.055$, $SD = 0.316$): mouth ($M = 0.101$, $SD = 0.116$), $t(26) = 3.24$, $p = .003$, $d_z = 0.62$, 95% CI [0.21 1.03]; right cheek ($M = 0.064$, $SD = 0.162$), $t(26) = 3.27$, $p = .003$, $d_z = 0.63$, 95% CI [0.21 1.04]; left cheek ($M = 0.137$, $SD = 0.107$), $t(26) = 3.17$, $p = .004$, $d_z = 0.62$, 95% CI [0.19 1.02].

For saccades from fixation at the mouth, there was no main effect of target location or of emotion on the directional strength of those saccades, nor an effect of the interaction (all $Fs < 1.6$, $ps > .2$).

For saccades from fixation at the left cheek, there was no main effect of target location or of emotion on the directional strength of those saccades (both $Fs < 1.2$, $ps > .3$). The effect of the interaction was small and not statistically significant, $F(2.71, 70.56) = 2.29$, $p = .091$, $\eta p^2 = .08$, 90% CI [NA .17], $\eta_G^2 = .009$.

For saccades from fixation at the right cheek, there was a significant main effect of target location, $F(1.06, 27.43) = 10.1$, $p = .003$, $\eta p^2 = .28$, 90% CI [.07 .46], $\eta_G^2 = .13$, but no main effect of emotion ($F < 1$, $p > .6$). Reflexive first saccades from the right cheek were most strongly directed towards the mouth ($M = 0.38$, $SD = 0.165$) followed by the left cheek ($M = 0.311$, $SD = 0.103$), left eye ($M = 0.275$, $SD = 0.94$), brow ($M = 0.258$, $SD = 0.102$), and right eye ($M = 0.242$, $SD = 0.131$). The results of the pairwise comparisons can be seen in S1 Table. There was a marginal effect of the interaction between emotion and target location, $F$

                                

$(2.9, 75.4) = 2.69$, $p = .054$, $\eta p^2 = .09$, 90% CI [NA .18], $\eta_G^2 = .006$. Simple main effects analyses revealed effects of target location for all 4 emotions (all $F$s > 5.5, $p$s < .001). Pairwise comparisons to follow up these simple main effects are reported in S2 Table. For angry faces, reflexive first saccades from the right cheek were directed more strongly towards the mouth ($M = 0.373$, $SD = 0.179$) compared to the left cheek ($M = 0.31$, $SD = 0.109$) and left eye ($M = 0.275$, $SD = 0.095$). For fearful faces, the most notable results were that reflexive first saccades from the right cheek were directed more strongly towards the mouth ($M = 0.411$, $SD = 0.184$) than towards all other target locations: right eye ($M = 0.231$, $SD = 0.147$), brow ($M = 0.258$, $SD = 0.11$), left eye ($M = 0.28$, $SD = 0.1$), and left cheek ($M = 0.326$, $SD = 0.111$). Similarly, for surprised faces, the most notable results were that reflexive first saccades were more strongly directed towards the mouth ($M = 0.383$, $SD = 0.169$) compared to all other target locations: right eye ($M = 0.235$, $SD = 0.164$), brow ($M = 0.248$, $SD = 0.124$), left eye ($M = 0.265$, $SD = 0.105$), and left cheek ($M = 0.304$, $SD = 0.103$). A similar pattern of results was also evident for sad faces, with the saccades more strongly in the direction of the mouth ($M = 0.354$, $SD = 0.164$) than of the right eye ($M = 0.257$, $SD = 0.124$), brow ($M = 0.266$, $SD = 0.107$), left eye ($M = 0.279$, $SD = 0.11$) and left cheek ($M = 0.303$, $SD = 0.12$); however, none of the pairwise comparisons was significant after correction for multiple comparisons (all uncorrected $p$s $\geq$ .012, $d_z$s $\leq$ 0.52; Bonferroni-Holm adjusted $\alpha = .005$).

## Discussion

Using a combination of angry, fearful, surprised and sad expressions in a brief-fixation paradigm, we aimed to investigate the contribution of initially fixating an informative facial feature to emotion recognition and seeking out of informative facial features when these are not initially fixated. We found an interaction between expression and initial fixation location on recognition accuracy. Initially fixating on the central brow of angry faces led to greater recognition accuracy compared to initially fixating on a cheek, as expected and as found by Atkinson and Smithson [1], though not compared to fixation on an eye or the mouth; indeed, fixation on the mouth also elicited greater anger recognition accuracy than fixation on a cheek. The greater accuracy for anger with fixation at the brow was associated with a reduction in the misclassifications of anger as sadness–this is akin to the reduction in the misclassifications of anger as neutral, as found by Atkinson and Smithson (2020), who used neutral but not sad faces. For fearful and surprised facial expressions, fixation on the central mouth led to better recognition, compared to fixation on an eye or the brow, which was associated with a reduction in the misclassifications between these expressions, as expected and as found by Atkinson and Smithson [1]. No effect of initial fixation location was found for sad faces.

Using our saccade path measure, we found that, when the data were collapsed across fixation location, reflexive first saccades were more strongly in the direction of the brow than of most or all the other locations of interest (left and right eyes, left and right cheeks, mouth). If reflexive first saccades target emotion-informative facial features, then we would have expected to find this result for angry faces and perhaps sad faces but not for fearful and surprised faces. Yet the brow had the largest normalized saccade path values for all 4 emotions. Moreover, if reflexive first saccades target emotion-informative facial features, then we would also have expected to find larger normalized saccade path values for the mouth than for other target locations for fearful and surprised and perhaps also sad faces, but no such effects were evident. Nonetheless, comparing the normalized saccade path values for the brow across emotions revealed that reflexive first saccades were more strongly directed towards the brow for angry faces and sad faces than for fearful faces, which hints at some small role for the emotion-informative nature of certain facial features in attracting saccades.

Analyses of the saccade path measures for the separate fixation locations revealed that the tendency for reflexive first saccades to be in the direction of the brow was evident only for saccades leaving the left and right eyes. Moreover, saccades leaving either the left or right eye were more strongly in the direction of the opposite eye than of the lower-face locations (mouth and cheeks). These findings help explain why there were proportionately fewer saccades downwards from the eyes compared to upwards from the mouth. The tendency for saccades to be directed towards the brow was not modified by the emotional expression when those saccades were initiated from the left eye, but it was for saccades initiated from the right eye. Reflexive first saccades from fixation at the right eye were directed more strongly towards the brow for angry faces than for fearful, surprised and sad faces. This is consistent with the hypothesis that emotion-informative facial features attract saccades, though note that, for saccades from the right eye as with saccades from the left eye, the brow was the location with the largest saccade path measures for all 4 emotions, not just for anger.

For reflexive first saccades from the brow itself, the saccade path measures were small with large variability, yet for angry faces, these values indicated a tendency for the direction of those saccades to be more towards the left cheek than the mouth and right cheek and more towards the mouth than the right cheek. This is reminiscent of Atkinson and Smithson's [1] finding of a bias towards the left and center of the face for downward saccades from the brow for angry, fearful, happy and neutral faces in their Experiment 1 and for angry and (less clearly) for fearful faces, but not for surprised or neutral faces, in their Experiment 2. For fearful faces in the present experiment, we found that there was a tendency for the saccades to be more in the direction of all 3 lower-face features than of the right (but not left) eye.

The other main notable findings from the saccade path analyses was that reflexive first saccades from the right cheek of fearful and surprised faces were directed more strongly towards the mouth than towards all other target locations. Although this might in part reflect a tendency for those saccades to target the closest facial feature to the fixated location, it is also consistent with the hypothesis that emotion-informative facial features–in this case, the mouth–attract saccades.

## Experiment 2

In Experiment 2a we aimed to address the same research questions as for Experiment 1, this time using a different combination of facial expressions. Those questions were: (1) Does a single fixation on an emotion-distinguishing facial feature enhance emotion-recognition accuracy? (2) Do reflexive first saccades from initial fixation on the face target emotion-distinguishing features? We replicated Experiment 1 using a new combination of facial expressions–angry, fearful, surprised and disgusted (replacing sad)–for two reasons. First, we wanted to replicate our results relating to anger, fear and surprise in the context of a different combination of emotions. Second, we wanted to test whether enforcing fixation on the mouth would enhance the recognition of disgust and whether reflexive saccades are more strongly directed to the mouth of disgusted faces, given previous research indicating that mid-to-high spatial frequency information at the mouth and the neighbouring wrinkled nose region is informative for the recognition of disgust [14–17]. Furthermore, previous research also suggests that observers spend more time looking at the mouth regions of disgusted facial expressions (more specifically, the upper lip in [40]). Previous research also suggests high confusion rates between disgust and anger, especially disgusted expressions being misclassified as angry [53–57]. Jack et al. [56] found that the resolution of this misclassification occurs when the upper lip raiser action unit is activated in the expression dynamics. Therefore, we hypothesized higher emotion recognition accuracy for disgusted faces and reduced misclassifications when the mouth is foveated.

In Experiment 2b we addressed two additional research questions: (3) Do observers, when required to classify facially expressed emotions, spend more time fixating the emotion-distinguishing facial features compared to less informative features? (4) Is the time spent fixating the emotion-distinguishing facial features related to accuracy in classifying the relevant emotion? Experiment 2b was the same as Experiment 2a with the principal exception that the face stimuli were displayed for considerably longer (5 s rather than 82 ms), thus allowing the participants to freely view the faces.

## Methods

**Participants.** Forty participants took part in the brief-fixation paradigm (Experiment 2a; female = 31, male = 9; mean age = 21.9 years, age range = 19–42) and of these 40, 39 participants also completed the long-presentation paradigm (Experiment 2b; female = 30, males = 9; mean age = 22 years, age range = 19–42). All participants were undergraduate or postgraduate students in Psychology and had normal or corrected-to-normal vision. All participants gave written consent to take part and undergraduate participants were rewarded participant pool credit (postgraduate participants did not receive any compensation for their time). The study was approved by the Durham University Psychology Department Ethics Sub-committee.

**Materials.** For Experiment 2a (brief fixation), the face stimuli were identical to those of Experiment 1 except that faces with disgusted expressions replaced the sad faces (for the same identities) from the same face database (i.e., 96 images, comprising 24 identities × 4 emotions: anger, fear, surprise, disgust). For Experiment 2b (long presentation free-viewing), a subset of the face image set used in the brief-fixation experiment was used, such that there were 12 facial identities (6 males, 6 females), each presented in each of 4 expressions (angry, fearful, surprised and disgusted) leading to a total of 48 images.

**Design and procedure.** The design and procedure of Experiment 2a were identical to those of Experiment 1. Experiment 2b had the same design and procedure except that there were 4 blocks of 48 (rather than 96) trials and each face was presented for 5 s (rather than for 82.4 ms). Thus, over the 4 blocks (total trials = 192) of Experiment 2b, each of the 48 images (12 identities × 4 emotions) was presented once at each of the 4 initial fixation locations (eyes, brow, cheeks, mouth), with the eye and the cheek fixation locations selected equally on the left and right. Within each block, faces were presented at each of the 4 initial fixation locations 12 times (3 per emotion). The order of image presentation was randomised for each participant within each block. Participants were asked to press the relevant keyboard button as soon as they were confident what emotion was shown on the face and the face image remained on the screen for 5 s regardless of when they made their response. The order of Experiments 2a and 2b was counterbalanced for each participant so that half of the participants completed the brief-fixation paradigm first and long-presentation paradigm later and the other half did the opposite. The average delay between the two sessions for all participants was 3.2 days.

**Data analysis.** The data analysis procedures for Experiment 2a were the same as for Experiment 1. For both Experiments 2a and 2b, unbiased hit rates were calculated following the removal of trials with RTs < 200ms. For Experiment 2a this led to the removal of 0.02% of all trials, and for Experiment 2b, the removal of 0.01% of all trials.

For Experiment 2b, we also calculated for each participant the mean total fixation duration for each of 4 regions of interest (ROIs) per emotion, that is, for each participant we summed the durations of all fixations in each ROI per trial, up to the point at which the participant pressed the response button (or, in the very small number of trials where participants responded after face offset, for the full 5 s face duration), and then calculated the average of these total fixation durations for each emotion for each participant. This allowed us to assess

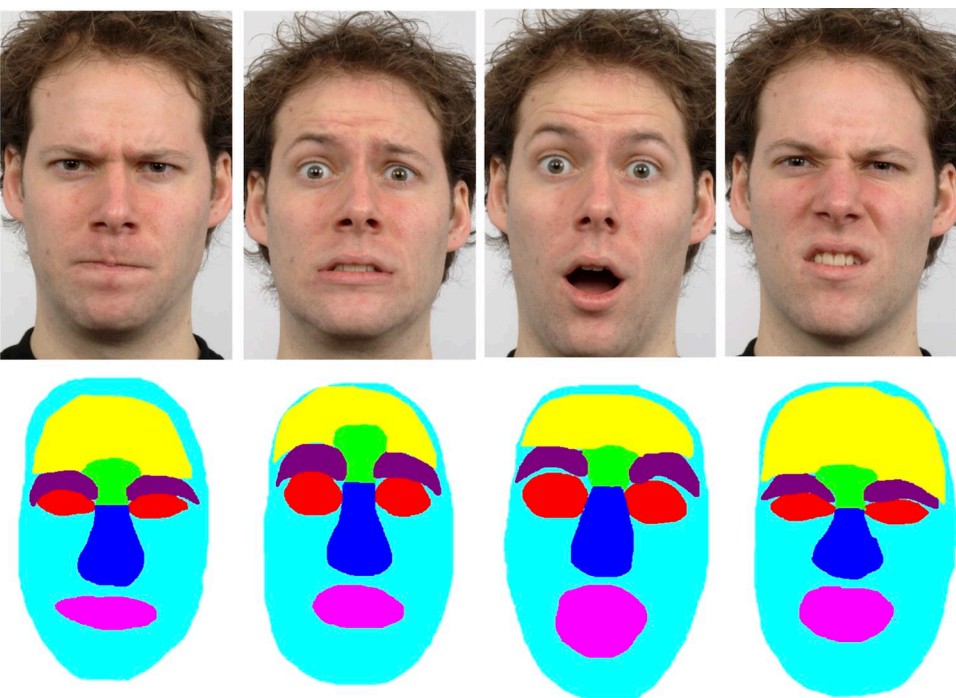

**Fig 4. Examples of facial expression images for Experiments 2a and 2b and corresponding ROIs.** From left to right: anger, fear, surprise, disgust. The size of each ROI is dependent on the underlying expressions and the shape of the facial feature; the forehead (yellow), eyebrows (deep purple) and rest of the face (cyan) regions were not included in the analysis of total fixation duration. The face images are republished in slightly adapted form from the Radboud Faces Database [45] under a CC BY license, with permission from Dr Gijsbert Bijlstra, Radboud University.

task-related fixations to facial regions of interest as a function of the displayed emotion and the relationship between those task-related fixations and emotion classification performance. (An alternative analysis, reported in the Supplementary Results and Discussion in S1 File, used the percentage of fixation times, relative to the total fixation duration on the image, rather than sum of fixation times. This controls for any differences in image viewing times across conditions and participants. The results of this alternative analysis were very similar to those reported here for the total fixation duration.)

The ROIs were the eyes (combining left and right eyes into a single ROI), brow, nose and mouth. These ROIs were drawn freehand, similar to some previous studies [e.g., 32,40,58], using a bespoke C++ programme (see Fig 4). This allowed us to delineate more precisely the shape and size of each ROI as compared to an alternative strategy of delineating rectangular ROIs [e.g., 37]; importantly, it also allowed for changes in the size of ROIs across emotions. Consequently, the average size of the ROIs varied and varied across emotions, as can be seen in Table 2. The average size of each ROI in degrees of visual angle is presented in Table 3. The

**Table 2. Mean sizes (pixels$^2$) of individually drawn ROIs.**

|  | Eyes | Brow | Nose | Mouth |
|---|---|---|---|---|
| **Angry** | 5949.8 | 4637.6 | 7716.2 | 6001.2 |
| **Fearful** | 8623.2 | 5304.9 | 8920.2 | 8286.8 |
| **Surprised** | 6774.7 | 4169.9 | 7667.8 | 8933.8 |
| **Disgusted** | 8545.5 | 4495.5 | 9067.4 | 9175.8 |

**Table 3. The average sizes of each ROI in degrees of visual angle (mean of maximum horizontal × maximum vertical) for each expression.**

|  | Left Eye | Brow | Right Eye | Nose | Mouth |
|---|---|---|---|---|---|
| Anger | 3.56 × 1.57 | 3.56 × 2.85 | 3.54 × 1.59 | 3.59 × 4.42 | 5.80 × 2.06 |
| Fear | 3.5 × 2.40 | 3.45 × 3.23 | 3.35 × 2.38 | 3.67 × 4.90 | 5.75 × 2.83 |
| Disgusted | 3.61 × 1.80 | 3.72 × 2.59 | 3.64 × 1.79 | 3.88 × 4.07 | 4.96 × 3.26 |
| Surprised | 3.49 × 2.41 | 3.36 × 2.72 | 3.42 × 2.31 | 3.60 × 5.10 | 4.76 × 3.72 |

typical average accuracy of the EyeLink 1000 system is 0.5˚, which is much smaller than the sizes of our chosen ROIs, giving us the confidence that any fixation falling within these ROIs will be captured accurately by the eye tracker. Several previous studies have controlled for variation in the size of ROIs by normalising the fixation measures relative to the areas of the relevant ROIs, to account for the possibility that larger ROIs would acquire more fixations [32,58–60]. This assumption that larger ROIs acquire more fixations has, however, been called into question [61], and is not supported by either our own data or that of some other groups using face stimuli [e.g., 40]. Our data shows that, averaged across emotions and initial fixation locations, the eyes received the most fixations per trial ($M = 3.4$, $SD = 1.9$), as compared to the nose ($M = 3.0$, $SD = 1.3$), mouth ($M = 1.2$, $SD = 0.8$) and brow ($M = 0.4$, $SD = 0.3$), and yet the eyes ROI was the third largest of the 4 ROIs on average. Moreover, variations in the sizes of each ROI across facial identities for a given emotion did not correlate with the number of fixations to those identity- and emotion-specific ROIs, with only one exception: the size of disgusted eyes across facial identities positively correlated with the mean number of fixations in those disgusted eyes, $r = .884$, $p < .001$ (all other $p$s > .05). Note that, as with the fixation duration data, these frequency data refer to fixations between face onset and the participant's button press (or, in the very small number of cases where participants responded after face offset, until the 5 s face offset). We therefore report and analyse the raw (i.e., non-normalised) fixation data.

## Results

**Experiment 2a: Brief-fixation paradigm.** *Emotion classification accuracy.* The unbiased hit rates are summarized in Fig 5A. An ANOVA on the arcsine square-root transformed unbiased hit rates revealed a main effect of emotion, $F(1.63, 63.54) = 6.96$, $p = .003$, $\eta p^2 = .15$, 90% CI [.03 .27], $\eta_G^2 = .04$. Emotion recognition accuracy for surprised expressions was significantly higher than the accuracy for fearful expressions ($p < .001$). There was also a main effect of fixation location, $F(3, 117) = 10.34$, $p = .003$, $\eta p^2 = .21$, 90% CI [.1 .3], $\eta_G^2 = .02$. Fixating on the mouth region led to higher emotion recognition accuracy compared to fixating on the eyes and the brow (both $p$s < .001) but there was no difference between the mouth and the cheek ($p = .52$). The interaction between emotion and initial fixation failed to reach significance ($F < 1$, $p > .8$).

To further investigate our hypotheses relating to each expression separately, planned comparisons were carried out. This resulted in 3 one-tailed paired samples t-tests for each expression (minimum Bonferroni-Holm adjusted α = .017).

Contrary to our prediction, classification accuracy for anger was not enhanced with fixation at the brow ($M = 0.615$, $SD = 0.215$) relative to any of the other 3 fixation locations: cheek ($M = 0.653$, $SD = 0.23$), mouth ($M = 0.671$, $SD = 0.237$) and eyes ($M = 0.61$, $SD = 0.191$) ($p$s > .3). Indeed, a full set of pairwise comparisons revealed greater accuracy for anger with fixation on the mouth or a cheek than on the brow: mouth > brow, $t(39) = -3.09$, $p = .004$, $d_z = -0.49$, 95% CI [-0.81–0.16]; cheek > brow, $t(39) = -2.53$, $p = .016$, $d_z = -0.4$, 95% CI [-0.72–0.08].

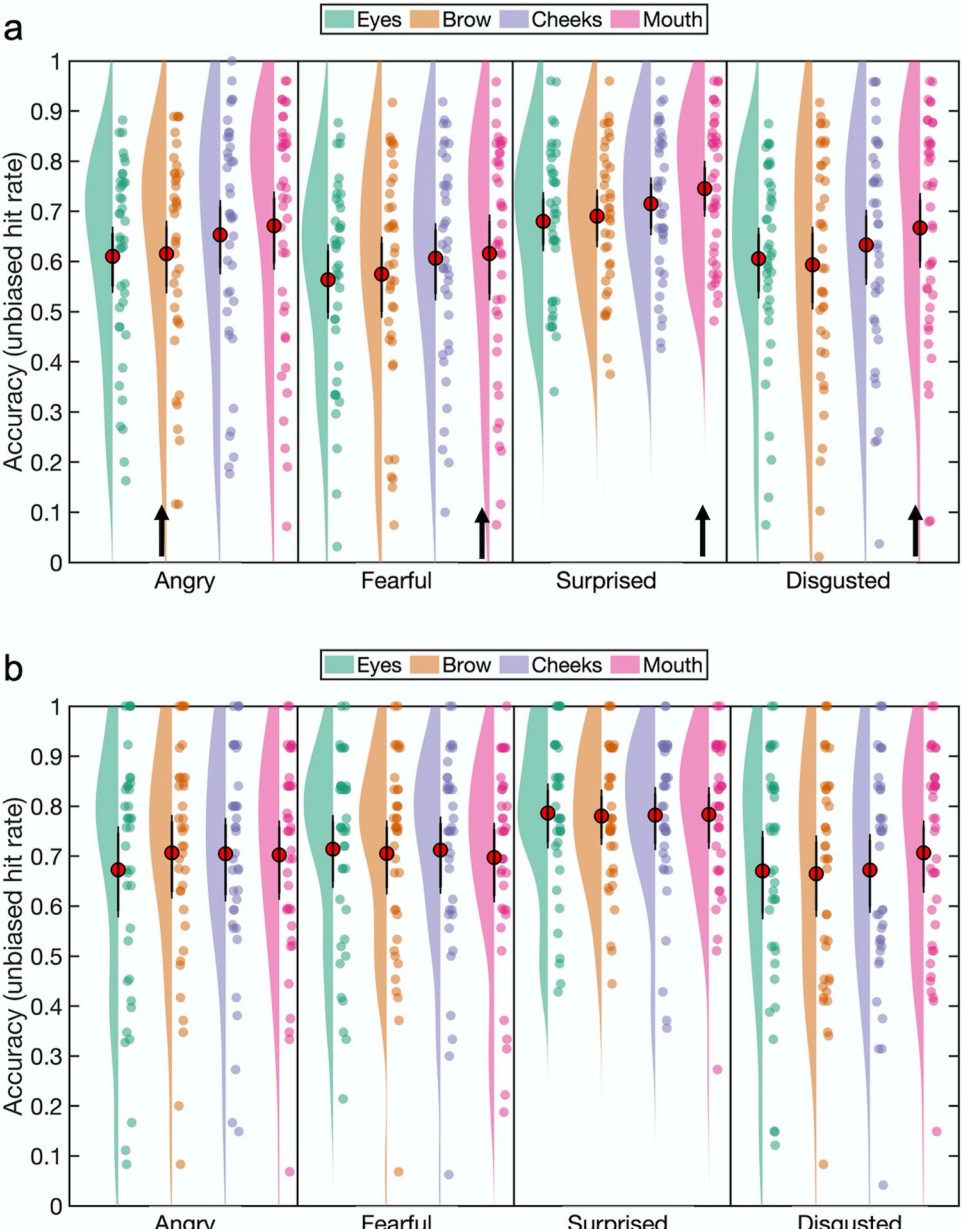

**Fig 5. Emotion recognition accuracy (mean unbiased hit rates) as a function of emotion category and fixation location.** Emotion recognition accuracy indexed by mean unbiased hit rates for the brief fixation paradigm in Experiment 2a (a) and the free viewing paradigm in Experiment 2b (b). Red circles indicate the mean value across participants and error bars indicate the 95% CIs (see Methods). The raincloud plot combines an illustration of data distribution (the 'cloud') with jittered individual participant means (the 'rain') for each condition [51]. Arrows indicate the most emotion-informative ('diagnostic') facial features for each emotion.

Anger classification accuracy was also greater with fixation on the mouth than on an eye $t(39) = 3.78$, $p < .001$, $d_z = 0.6$, 95% CI [0.26 0.93], and with fixation on a cheek than on an eye, $W(39) = 565$, $p = .005$, $d_z = 0.53$, 95% CI [0.22 0.74].

We expected that enforced fixation on the mouth would improve emotion recognition for fear, surprise and disgust. Accuracy for fear with fixation at the mouth ($M = 0.616$, $SD = 0.236$) was significantly higher compared to fixation on an eye ($M = 0.563$, $SD = 0.201$), $t(39) = 2.57$,

**Table 4. Confusion matrices for Experiment 2a (brief fixation paradigm).**

|  | Anger | Fear | Surprise | Disgust | Anger | Fear | Surprise | Disgust |
|---|---|---|---|---|---|---|---|---|
|  | Eyes | | | | Brow | | | |
| Anger | 79.58 | 6.77 | 1.15 | 12.50 | 80.83 | 5.00 | 2.40 | 11.77 |
| Fear | 3.02 | 69.69 | **22.08** | 5.21 | 3.23 | 68.33 | **23.96** | 4.48 |
| Surprise | 1.25 | 11.26 | 85.09 | 2.40 | 0.94 | 8.54 | 88.23 | 2.29 |
| Disgust | **23.88** | 0.83 | 1.67 | 73.62 | **26.15** | 1.35 | 2.08 | 70.42 |
|  | Cheeks | | | | Mouth | | | |
| Anger | 81.46 | 5.31 | 1.77 | 11.46 | 81.04 | 6.46 | 0.83 | 11.67 |
| Fear | 2.40 | 71.98 | **19.58** | 6.04 | 2.71 | 72.19 | **18.23** | 6.88 |
| Surprise | 0.63 | 9.49 | 88.01 | 1.88 | 0.94 | 7.60 | 90.10 | 1.35 |
| Disgust | **20.83** | 0.94 | 2.71 | 75.52 | **16.88** | 1.15 | 2.81 | 79.17 |

One confusion matrix is shown for each of the 4 initial fixation locations (eyes, brow, cheeks, mouth). The row labels indicate the presented expression and the column labels indicate the participant responses. The data are the % of trials each emotion category was given as the response to the presented expression. %s reported in bold represent the most prevalent confusions for each expression.

$p = .007$, $d_z = 0.41$, 95% CI [0.13 $\infty$], and with fixation at the brow ($M = 0.575$, $SD = 0.222$), though only marginally so after correction for multiple comparisons, $t(39) = 1.98$, $p = .028$, $d_z = 0.31$, 95% CI [0.04 $\infty$] (Bonferroni-Holm adjusted $\alpha = .025$), and not compared with fixation on a cheek ($M = 0.607$, $SD = 0.209$), $t < 1$, $p > .3$, $d_z = 0.08$.

Similar results were found for surprised faces: Recognition accuracy was higher with fixation at the mouth ($M = 0.745$, $SD = 0.142$) compared to the eyes ($M = 0.68$, $SD = 0.152$), $t(39) = 3.77$, $p < .001$, $d_z = 0.6$, 95% CI [0.31 $\infty$], brow ($M = 0.69$, $SD = 0.143$), $t(39) = 3.33$, $p < .001$, $d_z = 0.53$, 95% CI [0.25 $\infty$], and cheeks ($M = 0.715$, $SD = 0.145$), $t(39) = 1.94$, $p = .03$, $d_z = 0.31$, 95% CI [0.04 $\infty$].

Similar results were also found for disgust faces: Recognition accuracy was higher with fixation at the mouth ($M = 0.667$, $SD = 0.226$) compared to the eyes ($M = 0.605$, $SD = 0.197$), $t(39) = 2.98$, $p = .002$, $d_z = 0.47$, 95% CI [0.19 $\infty$], brow ($M = 0.593$, $SD = 0.235$), $t(39) = 3.55$, $p < .001$, $d_z = 0.56$, 95% CI [0.4228 $\infty$], and cheeks ($M = 0.633$, $SD = 0.223$), $t(39) = 1.97$, $p = .028$, $d_z = 0.31$, 95% CI [0.04 $\infty$].

To investigate whether there were any systematic misclassifications between the target expressions, we computed confusion matrices as seen in Table 4. Expressions of disgust were most often misclassified as anger, but this was greatly reduced with enforced brief fixation on the mouth. Fearful expressions were often misclassified as surprised, but this was reduced with enforced brief fixation on one of the lower facial features, i.e., a cheek or the mouth.

*Eye movement analysis.* As for Experiment 1, we next examined whether reflexive first saccades targeted expression-informative facial features, using the saccade path analysis described in the Data Analysis section. We first report an analysis with the data collapsed across fixation location, followed by separate analyses for each fixation location.

Reflexive saccade data from two participants were removed from the analysis of eye-movement data since they did not meet the criteria for inclusion (see Experiment 1). The following analyses were conducted on the data for 38 participants. On average, for the 38 participants, 82.74% of all trials included a reflexive saccade.

*Saccade path analysis: Collapsed across fixation location.* The mean saccade trajectories of the first saccades are shown in Fig 6A for each emotion, collapsed across initial fixation location; therefore, for each emotion there are 6 possible saccade targets (left and right eye, brow, left and right cheek, and the mouth). A 4 × 6 repeated measures ANOVA was run using emotion and saccade target location as factors. A marginal main effect of target location, $F(2.12,$

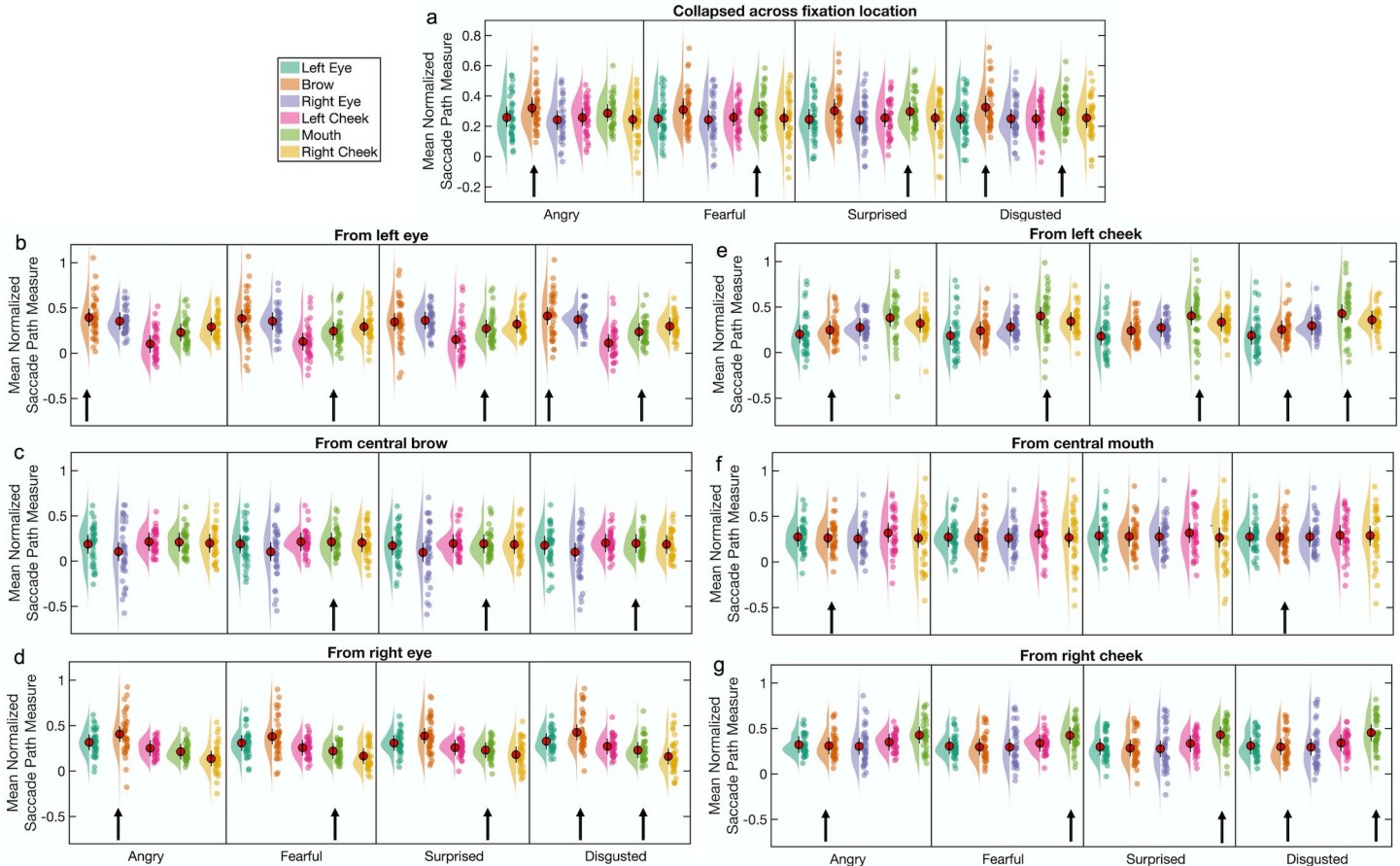

**Fig 6. Mean normalized saccade paths as a function of facial expression and target locations for Experiment 2a.** The normalized saccade path is a measure of the directional strength of the reflexive first saccades (executed after face offset) towards target locations of interest, in this case (a) to 6 target locations, collapsed across initial fixation location (N = 38), and from (b) the left eye, (c) the brow, (d) the right eye, (e) the left cheek, (f) the mouth, and (g) the right cheek, to the remaining 5 regions of interest (N = 38). Red circles indicate the mean value across participants and error bars indicate the 95% CIs (see Methods). The raincloud plot combines an illustration of data distribution (the 'cloud') with jittered individual participant means (the 'rain') for each condition [51]. Arrows indicate the most emotion-informative ('diagnostic') facial features for each emotion.

78.26) = 2.77, $p$ = .066, $\eta p^2$ = .07, 90% CI [NA .16], $\eta_G^2$ = .04, which indicated that first saccades were more strongly directed towards the brow ($M$ = 0.314, $SD$ = 0.135) than the left eye ($M$ = 0.25, $SD$ = 0.135), $t(37)$ = 4.29, $p$ < .001, $d_z$ = 0.7, 95% CI [0.34 1.05] and right eye ($M$ = 0.244, $SD$ = 0.135), $t(37)$ = 3.53, $p$ = .001, $d_z$ = 0.57, 95% CI [0.23 0.91], but not mouth ($M$ = 0.293, $SD$ = 0.112; $t$ < 1, $p$ > .5). First saccades also tended to be directed more strongly towards the brow than the left cheek ($M$ = 0.255, $SD$ = 0.11), $t(37)$ = 2.65, $p$ = .012, $d_z$ = 0.43, 95% CI [0.1 0.76] and right cheek ($M$ = 0.251, $SD$ = 0.14), $t(37)$ = 2.2, $p$ = .034, $d_z$ = 0.36, 95% CI [0.03 0.68], and more strongly towards the mouth than the right cheek, $t(37)$ = 2.25, $p$ = .03, $d_z$ = 0.37, 95% CI [0.03 0.69], though not after correction for multiple comparisons (adjusted $\alpha$ = .0033). There was no main effect of emotion ($F$ < 1, $p$ > .7) and a negligible effect of the interaction ($F$ = 1.51, $p$ = .15, $\eta p^2$ = .04, $\eta_G^2$ = .001). This analysis was also repeated with initial saccades that took place within 500ms of face offset, as reported in the Supplementary Results and Discussion in S1 File. In that analysis, no significant main effects or an interaction were found.

*Saccade path analysis*: *From fixation on individual features*. Separate 4 × 5 repeated measures ANOVAs were conducted for each of the 6 fixation locations, with emotion and saccade target as factors. The data are summarized in Fig 6B–6G.

For reflexive first saccades from fixation at the left eye, there was a main effect of target location, $F(1.1, 41.06) = 24.08$, $p < .001$, $\eta p^2 = .39$, 90% CI [.19 .53], $\eta_G^2 = .22$. The results of follow-up pairwise comparisons are shown in S3 Table. The interaction between expression and target location failed to reach significance after Greenhouse-Geisser correction, $F(3.85, 142.54) = 2.32$, $p = .062$, $\eta p^2 = .06$, 90% CI [NA .11], $\eta_G^2 = .008$. The main effect of emotion was not significant either ($F < 1$, $p > .5$).

For reflexive first saccades from fixation at the brow, there was no main effect of target location or of emotion, nor an effect of the interaction ($Fs < 2.4$, $ps > .1$, $\eta p^2s < .07$).

For reflexive first saccades from fixation at the right eye, there was a main effect of target location, $F(1.06, 39.07) = 26.22$, $p < .001$, $\eta p^2 = .42$, 90% CI [.21 .55], $\eta_G^2 = .24$. The results of follow-up pairwise comparisons are shown in S3 Table. The main effect of emotion and the effect of the interaction were negligible (both $Fs < 1.8$, $ps = .16$, $\eta p^2s \leq .05$).

The directional strength of reflexive first saccades from fixation on the mouth did not vary as a function of target location or emotion or their interaction ($Fs < 1.1$, $ps > .35$, $\eta p^2s < .03$). The directional strength of reflexive first saccades from fixation on the left cheek varied as a function of target location, $F(1.05, 38.76) = 10.04$, $p = .003$, $\eta p^2 = .21$, 90% CI [.05 .38], $\eta_G^2 = .14$, but not as a function of emotion or of the interaction ($Fs < 1.1$, $ps > .35$, $\eta p^2s < .03$). Similarly, the directional strength of reflexive first saccades from fixation on the right cheek varied as a function of target location, $F(1.04, 38.52) = 10.99$, $p = .002$, $\eta p^2 = .23$, 90% CI [.06 .39], $\eta_G^2 = .098$, but not as a function of emotion or of the interaction ($Fs < 1.1$, $ps > .35$, $\eta p^2s < .03$). The results of both sets of pairwise comparisons are shown in S3 Table. Similar to the analysis of saccades collapsed across initial fixation, the analysis of saccades starting within 500ms of face offset from each initial fixation location is reported in Supplementary Results and Discussions in S1 File. This analysis yielded similar results to those reported here.

## Experiment 2b: Long-presentation paradigm

*Emotion classification accuracy.* The descriptive statistics for the unbiased hit rates can be seen in Fig 5B. The ANOVA on the arcsine square-root transformed unbiased hit rates revealed a main effect of emotion, $F(1.53, 58.11) = 5.58$, $p = .011$, $\eta p^2 = .13$, 90% CI [.02 .25], $\eta_G^2 = .032$. Surprise ($M = 0.783$, $SD = 0.13$) was better recognised compared to fear ($M = 0.707$, $SD = 0.174$), $t(38) = 4.77$, $p < .001$, $d_z = 0.76$, 95% CI [.4 1.12] and disgust ($M = 0.678$, $SD = 0.196$), $t(38) = 3.24$, $p = .002$, $d_z = 0.52$, 95% CI [0.18 0.85]. There was no effect of fixation location and no significant interaction between emotion and fixation location ($Fs < 1$, $ps > .5$, $\eta p^2s < .03$).

To investigate whether there were any systematic misclassifications between target expressions, we computed confusion matrices as seen in Table 5. Similar to the brief-fixation paradigm, disgust was misclassified most often as anger and fear was often misclassified as surprise. Misclassification of disgust as anger was reduced with fixation on the mouth, whereas misclassification of fear as surprise was slightly reduced with fixation on the brow.

*Eye movement analysis: Mean total fixation duration.* To investigate whether participants spent more time fixating the informative facial features for each of the expressions in the experiment, we calculated the mean total fixation duration, up to the point of the participant's button press, for the eyes, brow, nose and mouth ROIs per emotion per participant. Descriptive statistics can be seen in Fig 7. A repeated measures ANOVA on these total fixation durations revealed significant main effects of emotion, $F(3, 114) = 70.74$, $p < .001$, $\eta p^2 = .651$, 90% CI [.56 .7], $\eta_G^2 = .007$, and region of interest, $F(1.7, 64.43) = 39.97$, $p < .001$, $\eta p^2 = .513$, 90% CI [.36 .61], $\eta_G^2 = .484$. The main effect of emotion reflected that the mean total fixation durations were longer for disgusted faces ($M = 724$ms, $SD = 110$) than for surprised ($M = 706$ms,

**Table 5. Confusion matrices for Experiment 2b (free-viewing paradigm).**

| | Anger | Fear | Surprise | Disgust | Anger | Fear | Surprise | Disgust |
|---|---|---|---|---|---|---|---|---|
| | | | Eyes | | | | Brow | |
| Anger | 81.76 | 5.58 | 1.72 | **10.94** | 85.65 | 3.64 | 0.86 | **9.85** |
| Fear | 1.07 | 79.23 | **15.63** | 4.07 | 1.29 | 78.76 | **13.52** | 6.44 |
| Surprise | 0.21 | 5.15 | 91.85 | 2.79 | 0.43 | 7.33 | 89.22 | 3.02 |
| Disgust | **20.86** | 0.22 | 0.22 | 78.71 | **20.47** | 0.43 | 0.22 | 78.88 |
| | | | Cheeks | | | | Mouth | |
| Anger | 85.26 | 4.70 | 0.85 | **9.19** | 84.05 | 5.17 | 0.86 | **9.91** |
| Fear | 1.72 | 78.88 | **15.09** | 4.31 | 1.50 | 78.33 | **15.02** | 5.15 |
| Surprise | 0.00 | 6.00 | 90.79 | 3.21 | 0.86 | 6.25 | 90.52 | 2.37 |
| Disgust | **19.78** | 0.86 | 1.08 | 78.28 | **17.63** | 0.00 | 0.00 | 82.37 |

One confusion matrix is shown for each of the 4 initial fixation locations (eyes, brow, cheeks, mouth). The row labels indicate the presented expression and the column labels indicate the participant responses. The data are the % of trials each emotion category was given as the response to the presented expression. %s reported in bold represent the most prevalent confusions for each expression.

$SD$ = 106; uncorrected $p$ = .004, $d_z$ = 0.487), fearful ($M$ = 697ms, $SD$ = 104; uncorrected $p$ = .002, $d_z$ = 0.52) and angry ($M$ = 622ms, $SD$ = 110; uncorrected $p < .001$, $d_z$ = 2.107) faces, longer for surprised (uncorrected $p < .001$, $d_z$ = 1.639) and fearful (uncorrected $p < .001$, $d_z$ = 1.691) than for angry faces, but did not differ between fearful and surprised faces (uncorrected $p > .2$, $d_z$ = 0.184). The main effect of ROI showed that the mean total fixation durations were longer for the eyes ($M$ = 1227ms, $SD$ = 605) than for the brow ($M$ = 125ms, $SD$ = 88; uncorrected $p < .001$, $d_z$ = 1.767) and mouth ($M$ = 420ms, $SD$ = 360; uncorrected $p < .001$, $d_z$ = 1.033) but not nose ($M$ = 977ms, $SD$ = 531; uncorrected $p > .1$, $d_z$ = 0.241), and longer for the nose than for the brow (uncorrected $p < .001$, $d_z$ = 1.636) and mouth (uncorrected $p < .001$, $d_z$ = 0.802), and longer for the mouth than for the brow (uncorrected $p < .001$, $d_z$ = 0.759).

There was also a significant Emotion × ROI interaction, $F(5.15, 195.85)$ = 32.81, $p < .001$, $\eta p^2$ = .463, 90% CI [.37 .52], $\eta_G^2$ = .043. Simple main effects analyses revealed significant effects of ROI for each of the 4 emotions ($Fs \geq 28.53$, $ps < .001$) as well as significant effects of emotion for each of the 4 ROIs ($Fs \geq 8.74$, $ps < .001$). Pairwise comparisons for the main effects of emotion are reported in S4 Table, which reveal several findings consistent with our hypothesis that observers will spend more time fixating emotion-distinguishing than less informative facial features. Notably, participants spent more time fixating (1) the eyes for fearful and

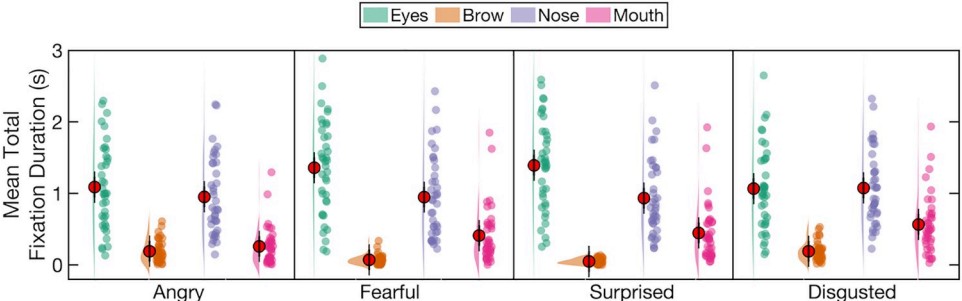

**Fig 7. The mean total fixation duration per trial for the four ROIs for each emotion in Experiment 2b.** *Note*: Red circles indicate the mean value across participants and error bars indicate the 95% CIs (see Methods). The raincloud plot combines an illustration of data distribution (the 'cloud') with jittered individual participant means (the 'rain') for each condition [51]. The ROIs are illustrated in Fig 4.

surprised faces than for angry and disgusted faces; (2) the brow for angry and disgusted faces than for fearful and surprised faces; (3) the nose for disgusted faces than for fearful, surprised and angry faces; and (4) the mouth for fearful and surprised faces than for angry faces and for disgusted faces than for angry, fearful and surprised faces. Additional pairwise comparisons comparing across ROIs for each emotion, reported in S5 Table, further revealed (5) that participants spent more time fixating the mouth than the brow for disgusted, fearful and surprised faces, but for angry faces fixation duration did not differ between the brow and the mouth. Given the importance of the eyes in fearful and surprised faces and their visual similarity across these two emotions, it is also interesting to note that there was a small but, after corrections for multiple comparisons, statistically non-significant tendency for participants to spend more time fixating the eyes than the nose for fearful and surprised faces but not for angry and disgusted faces. This analysis was repeated with percentage of total fixation duration relative to the total fixation duration on the image for each trial and the results remained the same.

*Relationship between total fixation duration and emotion recognition accuracy*. To investigate whether there is a relationship between time spent fixating a region of informative value for the emotion and emotion recognition accuracy in Experiment 2b, we calculated Spearman's correlations between the time spent fixating an informative region (mean total fixation duration) and recognition accuracy (unbiased hit rates). Given the results for Experiment 2a (the brief-fixation experiment) and for the fixation durations reported above, we focused these correlational analyses on the brow and mouth for angry faces, the brow, mouth and nose for disgusted faces, and the eyes and mouth for fearful and surprised faces. In each case, our hypothesis was that emotion classification accuracy would be positively correlated with time spent fixating the relevant region of interest.

Anger classification accuracy was positively correlated with the amount of time spent fixating the mouth, $\rho = .61$, $p < .001$ (Fig 8A), but not brow ($p = .47$) of angry faces. Disgust classification accuracy was positively correlated with the amount of time spent fixating the mouth, $\rho = .5$, $p < .001$ (Fig 8B), but not brow ($p = .44$) or nose ($p = .76$) of disgust faces. There was a small positive correlation between the amount of time spent fixating the mouth of fearful faces and classification accuracy for fear, $\rho = .32$, $p = .025$ (Fig 8C). Similarly, there was a small positive correlation between the amount of time spent fixating the mouth of surprised faces and classification accuracy for surprise, $\rho = .27$, $p = .046$ (Fig 8D). The amount of time spent fixating the eyes was not positively correlated with classification accuracy for either fear ($p = .92$) or surprise ($p = .83$). When considering percentage of total fixation duration relative to the total fixation duration on the image, the results were similar (all reported effects held).

In further exploratory analyses, we examined (1) relationships between time spent fixating the mouth or central brow of the different emotional faces in Experiment 2b and recognition accuracy with enforced brief fixation on those features in Experiment 2a, given that the same participants took part in both experiments, and (2) whether fixation duration in any of the 4 ROIs, regardless of the emotion expressed on the face, was related to accuracy in classifying any of the emotions. These analyses are reported in the Supplementary Results and Discussion in S1 File.

## Discussion

In the brief fixation paradigm (Experiment 2a), enforced fixation on the mouth or cheek of angry faces led to higher classification accuracy compared to enforced fixation on an eye or the brow. This finding contrasts with what we found in Experiment 1 and in a previous study [1], where accuracy for anger was higher with enforced brief fixation on the brow than on an eye or cheek. The present finding is nonetheless consistent with the informative nature of the

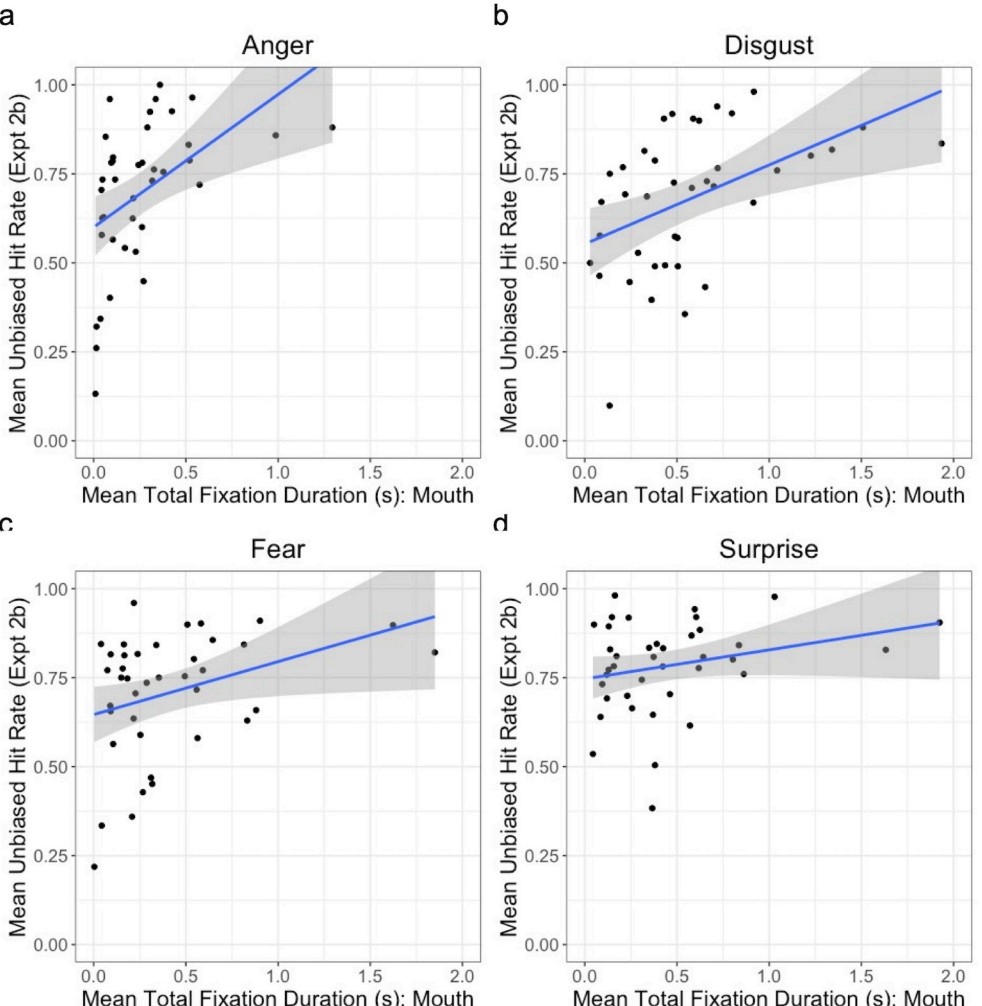

**Fig 8. Relationships between fixation duration on the mouth and emotion classification accuracy.** Panels show the associations between fixation duration on the mouth (a) angry, (b) disgusted, (c) fearful and (d) surprised faces in Experiment 2b and emotion classification accuracy for those same emotions in Experiment 2b (free viewing). Each dot represents a single participant. Shaded area indicates the 95% confidence interval.

mouth region in tasks requiring participants to distinguish angry from certain other emotional expressions [16,17]. Fixation on the mouth also led to higher recognition accuracy for fearful and surprised expressions relative to fixation of other locations (relative to the eyes and brow for fear and relative to the eyes, brow and cheeks for surprise), similar to what we found in Experiment 1 and in a previous study [1]. A novel finding in the present experiment was that enforced brief fixation on the mouth also led to higher recognition accuracy for disgusted expressions relative to fixation of an eye, the brow or a cheek. The enhanced classification of anger with fixation at the mouth or a cheek was associated with a relatively large reduction in the misclassifications of angry expressions as disgusted (especially with fixation at the mouth). The enhanced classification of disgust with fixation at the mouth was associated with a reduction in the misclassifications of disgusted expressions as angry. The enhanced classification of surprise and fear with fixation at the mouth was associated with a reduction in the confusions between these emotions, similar to Experiment 1 and our previous study [1].

Analyses of our saccade path measure showed that reflexive first saccades from enforced fixation on either the brow or the mouth did not selectively target any of the other locations of interest (left and right eyes and cheeks), whereas saccades from fixation on a left or right eye or cheek were directed more strongly towards the closest facial feature rather than the emotion-distinguishing features: saccades initiated from the left or right eye were directed towards the brow or the other eye, replicating a finding in Experiment 1, and saccades initiated from the left or right cheek were directed more towards the mouth or the other cheek. Moreover, analyses collapsed across initial fixation location showed that reflexive first saccades tended to be directed more strongly towards the brow than towards any of the other locations of interest. Importantly, however, none of these effects varied as a function of the emotional expression, as would be expected if reflexive first saccades targeted emotion-distinguishing features. Nonetheless, exploratory analyses of the saccade path measures for first saccades after face onset in Experiment 2b, in which the faces were presented for longer, did in fact provide evidence consistent with those initial saccades targeting emotion-informative facial regions (see Supplementary Results and Discussion in S1 File). Specifically, first saccades were more strongly in the direction of the central brow for angry compared to fearful, surprised and disgusted faces, and more strongly towards the mouth for surprised and disgusted faces than for angry or fearful faces.

When allowed to free-view faces (Experiment 2b), observers still misclassified angry faces as disgusted and vice versa, at approximately the same frequency as in Experiment 2a, despite the large difference in stimulus duration. They also misclassified fearful faces as surprised and vice versa, though less often than they did in Experiment 2a. As expected, initial fixation location had no effect on emotion classification accuracy in Experiment 2b.

Analyses of total fixation durations up to the point of the participant's button press in Experiment 2b showed that, although observers spent most of the time fixating the eyes followed closely by the nose, there were also some interesting differences in fixation durations across emotions for certain ROIs. Observers tended to fixate longer on emotion-relevant (if not always emotion-distinguishing) facial features: the eyes for fearful and surprised faces more than for angry and disgusted faces, the mouth for disgusted faces more than for fearful, surprised and angry faces and for fearful and surprised faces more than for angry faces, the brow more for angry and disgusted faces than for fearful and surprised faces, and the nose for disgusted faces more than for angry, fearful and surprised faces. These findings are in line with what we expected given the distribution of informative facial features for the particular combination of emotions used in this experiment. A comparison of these findings with those of previous studies is provided in the General Discussion. For now, we note two things. First, the fact that observers spent similar durations at the brow region of angry and disgusted faces might have contributed to the confusions between these two expressions, possibly due to their visual similarity (i.e., the furrows of the brow region). Second, no differences in the total fixation durations for any of the ROIs were found between the fearful and surprised facial expressions, which might have contributed to the confusion of these expressions since the observers might have neglected to fixate on the distinctive features of fear and surprise.

Finally, we found that time spent fixating the mouth in Experiment 2b was related to accuracy in classifying angry and disgusted expressions. Participants who spent more time fixating the mouth of angry and disgusted expressions in Experiment 2b were more accurate in classifying those expressions, not only in that same experiment but also in Experiment 2a in the condition in which the only feature they foveated was the mouth (and then only for 82 ms). Moreover, more time spent fixating the mouth in general, regardless of emotional expression, was associated with greater anger and disgust classification accuracy in both experiments. There was no relationship between brow total fixation duration in Experiment 2b and emotion

classification accuracy in either experiment for anger or disgust. Total fixation durations for neither the mouth nor the eyes were associated with classification accuracy for surprise in either experiment, however; there were similarly null findings for fear, except for a small positive association between total fixation duration on the mouth in Experiment 2b and fear classification accuracy in Experiment 2a.

## General discussion

Foveal and extrafoveal visual processing differ both quantitatively and qualitatively. At everyday interpersonal distances of ~ 0.5–2 m, another's face covers an area of one's visual field substantially larger than the area captured by the fovea [7]. Certain parts of the face and spatial frequencies at those locations are more informative than others for determining its emotional expression, though this can vary depending on what emotions the observer has to discriminate [14–17]. We tested four hypotheses derived from these facts: (1) A single fixation on an emotion-distinguishing facial feature will enhance emotion identification performance compared to when a less informative part of the face is fixated, especially when the emotion-distinguishing feature contains high spatial-frequencies. (2) Observers' reflexive first saccades from that single fixation on the face will preferentially target emotion-distinguishing facial features when those features are not already fixated. (3) When required to classify facially expressed emotions under free-viewing conditions, participants will spend more time fixating the emotion-distinguishing facial features compared to less informative features. (4) Greater emotion-classification accuracy will be associated with longer time spent fixating the emotion-distinguishing facial features. We found circumscribed support for hypotheses 1, 2 and 4, and only very limited support for hypothesis 2. In what follows, we discuss each of these in turn.

### Hypothesis 1: Foveating emotion-distinguishing facial features will enhance emotion recognition accuracy

In two brief-fixation experiments, we presented faces for a brief time (~82 ms), insufficient for a saccade, at a spatial position that guaranteed that a given feature–the left or right eye, the central brow, the left or right cheek, or the centre of the mouth–fell at the fovea and thus that the rest of the face was projected to the extrafoveal retina. In both these experiments, enforced fixation on the mouth resulted in better recognition for fearful and surprised expressions compared to fixation on the eyes and, with the exception of fear in the second experiment, on the brow [see also 1]. That fear was not best recognised with fixation on an eye is perhaps surprising, given that use of the Bubbles methodology has shown that high spatial frequency information from the eye region is informative for fearful expression recognition [14–17] and that the ability of a patient with bilateral amygdala damage to recognise fear in faces was restored when she was instructed to direct her gaze to the eye region [42]. Nonetheless, our finding here replicates a finding from our earlier study [1] and is consistent with previous work showing that the mouth principally distinguishes fearful from surprised as well as from neutral and angry expressions, whereas the eye region does not distinguish between prototypical fearful and surprised expressions [14,15,19,20]. Yet, under free-viewing conditions in the Bubbles task, it is the mid-to-low spatial-frequency information (7.5–60 cycles per image) at the mouth that observers tend to rely on most for surprise, at least when required to distinguish surprise from the other 5 basic emotions and neutral [14,15], and which they also use for distinguishing fearful from neutral faces alone, from angry and neutral faces, and from happy and neutral faces [17]. Thus, the improvement in the ability to distinguish between fearful and surprised faces when the mouth is foveated is unlikely to be entirely due to the additional high-resolution information extracted at fixation.

Enforced fixation on the mouth (and on a cheek) of angry faces also led to higher recognition accuracy for that emotion, relative to fixation of an eye or the brow (Experiment 2a). Yet in Experiment 1 and in a previous study [1], accuracy for anger was higher with enforced brief fixation on the brow than on an eye or cheek. The latter finding is consistent with findings of studies using the Bubbles technique showing that observers rely on the brow region, especially at mid-to-high spatial frequencies (15–60 cycles per image), to discriminate angry expressions from other emotional and emotionally neutral expressions [14–17]. The finding of enhanced anger recognition with fixation at the mouth but not with fixation at the brow in Experiment 2a was unexpected, but is nevertheless consistent with the informative nature of the mouth region in tasks requiring participants to distinguish angry from certain other emotional expressions, albeit not at the highest spatial frequencies [16,17]. Indeed, in Experiment 2a all the angry expressions had closed mouths and the surprised, disgusted and fearful expressions had more open mouths, whereas in Experiment 1 all the sad as well as the angry expressions had closed mouths; the distinctiveness of the closed mouth for angry expressions in Experiment 2a might thus have contributed to more accurate classification of anger with brief fixation on the mouth. These contrasts between Experiments 1 and 2a illustrate how the extent to which certain facial features are informative of the expressed emotion can vary depending on the combination of expressions used in the classification task [17].

Enforced fixation on the mouth of disgusted faces also led to higher recognition accuracy for that emotion, relative to fixation of an eye or the brow (Experiment 2a). This is consistent with the findings of studies showing that observers rely on the mouth and sides of the nose, especially at the middle to the highest spatial frequencies (15–120 cycles per image), to distinguish disgusted expressions from other emotional and emotionally neutral expressions [14,15,17,62]. Although enforced fixation on the mouth benefitted the recognition of disgust compared to enforced fixation on the upper facial features (i.e., the eyes and the brow), we did not find any difference in recognition accuracy for disgust with enforced fixation on the mouth compared to the cheeks. Additionally, although enforced fixation on the mouth reduced the confusion of disgust as anger, enforced fixation on one or other cheek also reduced this confusion albeit to a lesser degree. Although we chose the cheek region to be a relatively non-informative facial feature, it is possible that the chosen cheek location in this study is not as uninformative as expected. The location of the cheek positions is close to the nasolabial folds, which have been shown to be informative for disgust [15].

We did not find any differences in emotion classification accuracy for sad faces as a function of fixation location, despite the demonstrated use by observers of the brow and mouth regions, at both the highest (60–120 cycles per image) and mid-range (15–60 cycles per image) spatial frequencies, when required to distinguish sad faces from 5 other basic emotions and neutral [14,15]. Perhaps if sad expressions had been included in a stimulus set containing expressions with which sad faces are most typically confused–that is, with emotionally neutral, disgusted and fearful expressions [53,55,63–65]–then we might have found enhanced sadness recognition with fixation at the mouth or brow. Future work could test this prediction.

The enhanced accuracy for surprise and fear with fixation at the mouth was associated with a reduction in the confusions between these emotions [see also 1]. The confusion between fearful and surprised expressions is a prevalent finding in the literature, with fear typically being confused for surprise more often than vice versa [20,37,53,55,56,63,64,66]. This same asymmetry in confusion rates was also evident in our data; moreover, foveation of the mouth reduced misclassifications of fearful expressions as surprised much more than it reduced misclassification of surprised expressions as fearful.

In Experiment 1, the enhanced accuracy for angry faces with fixation at the brow was associated with a reduction in the misclassifications of angry faces as sad, whereas in Experiment

2a, the enhanced accuracy for angry faces with fixation at the mouth or a cheek was associated with a relatively large reduction in the misclassifications of angry expressions as disgusted (especially with fixation at the mouth).

The enhanced classification of disgust with fixation at the mouth was associated with a reduction in the misclassifications of disgusted expressions as angry. These two expressions are commonly confused, with anger typically being confused for disgust more often than vice versa [53,55–57,63,65]. This same asymmetry in confusion rates was also evident in our data; moreover, foveation of the mouth reduced misclassifications of disgusted expressions as angry whereas it did not reduce misclassifications of angry expressions as disgusted. Our findings here are also in line with work indicating that sampling information from visually similar facial features between anger and disgust, such as the eyes and eyebrows, at the expense of the distinctive facial features, such as the mouth, can lead to the confusion between these expressions [55]. On the other hand, in contrast to our findings, Poncet et al. [41] found the mouth region to be associated with increased misclassification of anger. One potential reason for the contradictory results might be a difference in the AUs used by the models in the two studies. While the AUs used by the models in the present experiments for angry expressions included the lip tightener and the lip presser, models in [41] used a variety of AUs leading to an open mouth for angry expressions. Since the open mouth is a shared feature in angry and disgusted expression, this might have led to confusions between these expressions while in our study the most distinctive feature between anger and disgust was the open mouth for disgust and the pressed lips for anger.

## Hypothesis 2: Reflexive first saccades will target emotion-distinguishing facial features

To gain insight to the potential targets of the reflexive first saccades in the brief-fixation experiments, we measured the similarity of the first saccade vector to six possible saccade vectors which targeted one of the possible initial fixation locations (i.e., left or right eye, brow, left or right cheek, mouth) on the face. This analysis showed that the saccades initiated at the left and right eyes were more strongly directed towards one of the other upper facial features (i.e., the opposite eye or the brow). This might be the reason behind the observation of lower proportion of saccades going downwards from the eyes, as reported in the Supplementary Results and Discussion in S1 File. When we consider the upper and lower portions of faces, there are more facial features of interest in the upper part of the face (i.e., two eyes, eyebrows, wrinkles of the brow) compared to the lower part (i.e., mouth). Additionally, the eye region contains more details in high spatial frequencies [e.g., 67]. Therefore, it is possible that observers examine the upper half of the face longer than the lower part simply due to the quantity of information within this region. This suggestion can be supported by our finding of longer average total fixation duration on the eyes compared to other regions of interest as well the finding of a higher percentage of reflexive saccades to the upper face from enforced fixation on the mouth compared to the percentage of saccades downwards from enforced fixation on an eye, as reported in the Supplementary Results and Discussion in S1 File.

Furthermore, detailed examination of the reflexive saccades originating from the cheeks also show that these saccades are directed towards a lower feature (i.e., opposite cheek and mouth) than one of the upper features. Arizpe et al. [68] showed that initial fixation location affected the target of subsequent saccades and showed that first fixations target the centre of the face, with a bias towards the location of the initial fixation so they end up close to the initial fixation location. When Scheller et al. [26] shifted the centre of the face images to the upper and lower halves of the screen they found that reflexive saccades targeted the facial feature that

was closest to the initial fixation location–in other words, when faces were presented in the upper half of the screen, initial fixations starting from the centre of the screen targeted the mouth region more compared to the eyes and vice versa. The initial saccades in both our experiments might be targeting the facial features that are spatially closest to the initially fixated location rather than being guided by the expression-informative facial features. Additionally, there is some evidence from our results, reported in the Supplementary Results and Discussion in S1 File, that the reflexive saccades are targeting the centre of the face in line with the centre-of-gravity effect and with the findings of our previous study using the brief fixation paradigm [1]. Previous research suggested that early fixations (within 0–250 ms after face onset) target the geometric centre of faces [32]. The reflexive saccades in our studies started less than 1000 ms after the onset of the face stimuli, and therefore it is possible that there is a mixture of early saccades that target the centre of the face and later saccades that might target expression-informative facial features among the reflexive saccades analysed. One important thing to note about the measurement of the centre-of-gravity effect reported in the Supplementary Results and Discussion in S1 File is that we used the nose region as the centre of the face. However, since the nose region covers a large area in the middle of the face, caution should be taken while interpreting the effect as the pure centre-of-gravity effect.

In a further analysis, we examined the paths of the first saccades after face onset in the long presentation paradigm of Experiment 2b (reported in the Supplementary Results and Discussion in S1 File) to more directly assess whether saccades target emotion-informative features. Contrary to the reflexive saccade results from the brief fixation paradigm, these saccades support the hypothesis that first saccades target emotion-informative facial features. More specifically, first saccades after face onset were more strongly in the direction of the central brow for angry compared to fearful, surprised and disgusted faces, and more strongly towards the mouth for surprised and disgusted faces than for angry or fearful faces. Therefore, it is possible that medium to low spatial frequency information from the emotion-informative facial features might guide the selection of the next saccade target as long as they are visually available.

## Hypotheses 3 and 4: Observers will spend more time fixating emotion-distinguishing than less informative facial features, which will be associated with greater emotion-classification accuracy

In Experiment 2b, the same participants who took part in Experiment 2a carried out a free-viewing task for the same expressions where they were allowed to examine the facial images for five seconds. This way, the observers were able to sample visual information from any region of the face freely. Previous research has shown that the eye (or more generally, the upper-face) region is the most frequently visited and longest viewed facial feature, but that the eye movements are still affected by the expressive content of the face [26,37–40,69,70]. Our results were consistent with this general pattern whilst also providing some novel insights.

Observers spent more time fixating the eyes for fearful and surprised faces than for angry and disgusted faces. This is consistent with the visual similarity of the eyes across these two emotions [14,15,19,20] and with the eyes being particularly informative for the recognition of fear [14–17]. Sullivan et al. [69] and Guo [39] also reported that younger adults (similar in age to our own participants) dwelled longer on the eyes of fearful and surprised faces than on the eyes of disgusted faces, though not compared to the eyes of angry faces. (In Sullivan et al.'s study, older adults actually dwelled longer on the eyes of angry than of fearful, surprised and disgusted faces.) In related findings, Schurgin et al. [40] reported longer total fixation durations for the eyes of fearful and angry faces, and shorter total fixation durations for the eyes of disgusted faces, compared to mean total fixation durations for the faces irrespective of

emotional expression. Both Sullivan et al. [69] and Guo [39] investigated all six expressions while we only used four. Among these expressions, anger and disgust are commonly confused, possibly partly due to the similarity of the eye and brow regions. Supporting this idea, initial fixation on the upper features (eyes and the brow) led to higher confusion of disgust as anger compared to lower features. Additionally, the results from Experiment 2a and 2b indicate that fixation on the mouth contributed to and was positively correlated with successful anger recognition. Therefore, it is possible that to avoid confusion between these visually similar expressions, participants looked at the eye region less in this study compared to studies where all six expressions are used.

Observers spent more time fixating the central brow region of angry and disgusted faces than of fearful and surprised faces. To the best of our knowledge, this is a novel finding, mainly because previous studies did not pick the central brow as a region of interest. Indeed, the bottom portion of our brow ROI is typically included in the eye region ROIs in those previous studies [37,38,44]. One study that chose the nasion region as a ROI [40] did not find that observers spent more time fixating this region for angry faces; however, they did find that observers spent more time looking at the nasion of neutral faces when these were presented in the same block with angry faces. This suggests that the nasion-brow region is important for discriminating angry from neutral faces, which is consistent with our previously published finding that enforced brief fixation on the central brow enhanced recognition of angry expressions and was associated with a reduction in the misclassifications of angry faces as neutral [1]. In the present experiments, we show that observers spent more time looking at the brow of angry (and disgusted) faces in the absence of neutral faces. Even though our brow ROI and Schurgin et al.'s nasion region might not be identical, they are partially overlapping. Our brow ROI encompasses the glabella which is above the nasion. There was also a task difference between our and Schurgin et al.'s studies. Where Schurgin et al. asked the participants to decide whether faces showed any expression, we asked our participants to explicitly choose an expression label for the faces they saw. Either one of these slight differences or a combination of them might have led to the differences between the two studies.

Observers spent more time fixating the nose for disgusted faces than for fearful, surprised and angry faces. Guo [39] similarly found longer dwell times and more fixations at the nose for disgusted faces compared to several other emotional expressions. Schurgin et al. [40] did not find longer fixation durations for the nose of disgusted faces, though they used two separate (upper and lower) nose ROIs and they did find that participants spent more time fixating the neighbouring upper lip for disgusted than for angry and sad faces. Our observers also fixated for longer on the mouth of disgusted compared to angry, fearful and surprised faces, and on the mouth of fearful and surprised compared to angry faces. These results appear to reflect the distribution of emotion-informative faces features for the particular combination of emotions used in this experiment. Specifically: the wrinkles around the lower nose and the mouth are particularly informative for the recognition of disgust activation of the upper lip raiser resolves the confusion of disgust as anger [56]; and the mouth differentiates surprise from fear [14–17].

In summary, consistent with previous findings, we show that eye movements are modulated by the emotion-informative facial features; however, taking into account research which suggests that visual information can be extracted from a single fixation near the nose [36,71,72] and that the location of fixations might not be perfectly linked to what information is being extracted [68,73], we also investigated whether there was a relationship between how long observers looked at the regions of interest and their recognition performance.

Consistent with the idea that time spent fixating certain facial features reflects the importance of those features for classifying the emotional expression, we found that time spent

fixating the mouth was positively correlated with accuracy in classifying disgusted and angry expressions. Consistent with the first of these results, Wong, Cronin-Golomb and Neargarder [70] found that older adults (mean age 65.9 years) who fixated the lower face more frequently than the upper face (or showed less of a preference for fixating the upper over the lower face) tended to be more accurate in classifying facial expressions of disgust. However, this correlation was not significant (even if in the same direction) for Wong et al.'s younger adult group, whose ages (mean 19.2 years) were more comparable to those of our own participants; moreover, Wong et al. found that anger classification accuracy was positively correlated with a fixation preference for the upper relative to the lower face in both the younger and older adults, which is not consistent with our finding that time spent fixating the mouth was positively correlated with accuracy in classifying angry expressions. Our finding for angry expressions may be explained by the subset of expressions we used. Angry expressions were the only expressions with a closed mouth, which might have become a cue for participants to discriminate this expression from fear, surprise and disgust.

We found that time spent fixating the mouth was not correlated with accuracy in classifying fearful and surprised expressions, as one might expect given previous research demonstrating the importance of the mouth in allowing observers to distinguish between these emotions (as discussed above) and as demonstrated in our brief-fixation experiments. It is possible that the mouth is more informative for fear recognition in short exposure durations, in which the ability to integrate information from multiple facial features is more restricted. We also found that time spent fixating the brow was not related to emotion classification accuracy for anger, despite previous evidence of the brow being informative for disgust recognition [1,14–17]; nor was time spent fixating the brow related to classification accuracy for disgust, fear or surprise; similarly, time spent fixating the eyes was not related to emotion classification accuracy for any of the 4 emotions.

Interestingly, time spent fixating the mouth of angry and disgusted expressions was positively correlated with accuracy in classifying those expressions also in Experiment 2a in the condition in which the only feature they foveated was the mouth (and then only for 82 ms) (as reported in the Supplementary Results and Discussion in S1 File). If we consider the eye movement strategies employed when the facial expression was presented for 5 seconds as the observer's idiosyncratic eye movement strategy, we can argue that participants who preferred to fixate the mouth region also benefited more from a forced fixation on this region. Previous research showed that observers have idiosyncratic eye movements which are stable across time and task [74–76]. Peterson and Eckstein [75] found that there are differences in the vertical location of first fixations among individual participants with some preferring upper face and some lower face locations. Arizpe et al. [76] further found that the spatial location of fixations cluster around four regions: left eye, right eye, the region between the eyes (nasion) and around the top of the nose/upper lip. Most significantly, Peterson and Eckstein [75] showed that these idiosyncratic eye movement preferences were functionally related to face identification performance, with the performance of those preferring upper face locations declining when forced to fixate lower facial features (i.e., tip of nose and mouth).

Given that we have used expressions posed at their peak intensity, emotion recognition performance was high for most of the studied expressions, even in the brief-fixation experiments. This might have masked the contribution of foveal processing of the respective informative features. Compare this with Vaidya et al. [44], for example, who showed (in a free-viewing paradigm) that fixations on different facial regions can be predictive of emotion recognition performance when the expressions presented are subtle but not extreme. In our experiments, it is possible that the peak intensity of the presented expressions might have reduced or even

eliminated the necessity of foveal processing of emotion-distinguishing features. In future work, we will address these outstanding issues using subtler emotional expressions.

Certain limitations of the experiments imposed by the participant demographics need to be acknowledged. Most of the participants in both studies were female. Previous research suggested a female superiority in emotion recognition [77–81] which was sometimes only reflected in faster RTs rather than accuracy [82,83]. Additionally, previous research suggests that females spend longer looking at the eye regions compared to males [78]. This female superiority is suggested to be most prominent for negative emotions [80,82]. However, while the literature on gender differences in emotion recognition is vast, the results are not always unequivocal. Some recent studies found no difference in the emotion recognition performance or eye movement patterns between young males and females [41,77,83] whose ages were close to the age of our sample. These groups of researchers suggest that that female superiority in emotion recognition is more prominent in older participants with older females better at recognizing emotions compared to older men and spending more time looking at the eye region more compared to older men. However, despite the equivocal results in the literature regarding gender differences in emotion recognition performance, it must be noted that the predominantly female sample in the experiments reported here might mean that the emotion recognition accuracy was higher than what can be expected in a more diverse population. Additionally, inclusion of more male participants might decrease the suggested informativeness of the eye region since males are suggested to look less at the eye region of faces.

The findings reported here can inform the study of facial expression perception in several ways. While fixations to the eyes and nose dominate the visual sampling of faces and facial expressions, this is more likely to be a face-specific strategy which applies to the majority of face-related tasks. The fixations that are of functional value–contributory to the task of emotion recognition–are the ones landing on distinctive facial features such as the mouth and the brow. The reflexive saccades that were previously suggested to be seeking-out informative features of facial expressions more likely reflect an automatic and involuntary strategy of looking upwards/at the upper visual field/upper face due to a top-down knowledge of the face configuration and are not best placed to measure what facial features are sought out when recognizing an expression. Combined with the previous findings that there are differences in the preferred fixation strategies among observers, our findings from the brief fixation paradigm suggest that observers with a preference to sample information from informative facial regions would perform better at emotion recognition. For example, those observers who prefer to fixate the lower face would perform better in recognition of all expressions studied here, especially when exposure duration is limited. This idea is further supported for angry and disgusted expressions since observers who freely looked longer at the mouth showed better anger and disgust recognition when briefly fixating the mouth of these expressions.

## Supporting information

**S1 Fig. The percentage of reflexive saccades going downwards from the eyes and upwards from the mouth for Experiment 1 and 2a.** We compared the percentage of first saccades that were directed upwards from the mouth to the percentage of first saccades downwards from the eyes. Percentages for each emotion were calculated relative to the total number of first saccades (up or down) from fixation on the eyes and mouth combined. The percentage of saccades going upwards from the mouth was significantly higher than the percentage of saccades going downwards from the eyes for both Experiment 1 (a) and Experiment 2a (b). The interaction between emotion and saccade direction indicated that the percentage of saccades going downwards from the eyes was lower for sad faces compared to surprised faces for Experiment

1. The main effect of emotion for percentage of saccades going downwards from the eyes indicated that there were fewer saccades leaving the eyes for angry expressions compared to fearful and surprised expressions. In the figure, the median percentage is represented by the middle horizontal line and the notch on each boxplot. The upper and lower horizontal lines of each box delineate the interquartile range (upper line represents the 75th percentile and lower line represents the 25th percentile). The percentages of reflexive saccades for each participant are overlaid on top of the boxplot to represent the distribution of the data and outliers. (TIF)

**S2 Fig. The percentage of reflexive saccades going downwards from the upper features and upwards from the lower features for Experiments 1 and 2a.** We compared the percentage of first saccades that were directed upwards from the lower facial features combined (cheeks + mouth) to the percentage of first saccades downwards from the upper facial features combined (eyes + brow). Percentages for each emotion were calculated as a percentage of the total number of initial saccades for each emotion per participant. The percentage of saccades going upwards from the lower features was significantly higher than the percentage of saccades going downwards from the upper features for both Experiment 1 (a) and Experiment 2a (b). Only in Experiment 2a, there were fewer saccades going downwards from upper features for angry faces compared to fearful faces. In the figure, the median percentage is represented by the middle horizontal line and the notch on each boxplot. The upper and lower horizontal lines of each box delineate the interquartile range (upper line represents the 75th percentile and lower line represents the 25th percentile). The percentages of reflexive saccades for each participant are overlaid on top of the boxplot to represent the distribution of the data and outliers. (TIF)

**S3 Fig. Centre-of-gravity effect indexed by the mean frequency of first saccades ending in the eye, brow, nose and cheek ROIs for each emotion for Experiments 1 and 2a.** To investigate whether participants demonstrated a tendency to direct their fixations towards the centre of faces, we compared the mean frequency of first saccades ending in the eye, brow, nose and cheek regions of interest for each emotion. For the purposes of this analysis, we will accept the nose region as the centre of the face however it should be noted that the definition of the nose in this study comprises the area between the bridge and apex of the nose. We found that the first saccades ended in the nose region significantly more frequently than in the eyes, brow, or the mouth both in Experiments 1 (A) and 2a (B) indicating that the first saccades were somewhat affected by the centre-of-gravity effect. In the figure, the median percentage is represented by the middle horizontal line and the notch on each boxplot. The upper and lower horizontal lines of each box delineate the interquartile range (upper line represents the 75th percentile and lower line represents the 25th percentile). The percentages of reflexive saccades for each participant are overlaid on top of the boxplot to represent the distribution of the data and outliers. (TIF)

**S4 Fig. The latency of reflexive saccades from each initial fixation location for Experiments 1 and 2a.** We investigated whether the initial fixation locations involved in this experiment influenced the latencies of reflexive first saccades. We found that the saccade latencies from initial fixation on the cheeks were shorter compared to all the other initial fixation locations for both Experiment 1 (A) and Experiment 2a (B). This indicates that the reflexive saccades in our study are influenced by the centre-of-gravity effect suggested by Bindemann et al. [32]. Only in Experiment 2a, the reflexive saccades from the eyes were shorter compared to reflexive

saccades from the brow. Additionally, for Experiment 2a, we find that first saccade latencies from the brow for angry faces were shorter compared to disgusted faces and saccade latencies from the cheeks were longer for fear compared to anger and disgust faces. In the figure, the median percentage is represented by the middle horizontal line and the notch on each boxplot. The upper and lower horizontal lines of each box delineate the interquartile range (upper line represents the 75th percentile and lower line represents the 25th percentile). The percentages of reflexive saccades for each participant are overlaid on top of the boxplot to represent the distribution of the data and outliers.
(TIF)

**S5 Fig. Relationships between fixation duration on the mouth in Experiment 2b and emotion classification accuracy in Experiment 2a.** Panels show the associations between fixation duration on the mouth for (a) angry and (b) disgusted faces in Experiment 2b and emotion classification accuracy for those same emotions in Experiment 2a (brief fixation). Each dot represents a single participant. Shaded area indicates the 95% confidence interval. We found positive correlations between fixation duration on the mouth in Experiment 2b and emotion classification accuracy for angry and disgusted expressions when fixation was enforced on the mouth in Experiment 2a.
(TIF)

**S6 Fig. Relationships between fixation duration on the mouth in Experiment 2b and emotion classification accuracy in Experiments 2a and 2b.** Panels show the associations between overall fixation duration on the mouth, regardless of emotion, in Experiment 2b and emotion classification accuracy for angry faces in (a) Experiment 2b (free viewing) and (b) Experiment 2a (brief fixation), and for disgusted faces in (c) Experiment 2b and (d) Experiment 2a, and for fearful faces in (e) Experiment 2b. Each dot represents a single participant. Shaded area indicates the 95% confidence interval. We found that fixating the mouth longer regardless of expression in Experiment 2b was positively correlated with anger and disgust classification accuracy in both Experiments 2a and 2b regardless of initial fixation location. Fixating the mouth longer in Experiment 2b was also marginally positively correlated with fear classification accuracy in Experiment 2b alone.
(TIF)

**S1 Table. Results of pairwise comparisons for the saccade path analyses for Experiment 1: Main effect of target location.**
(PDF)

**S2 Table. Results of pairwise comparisons for the saccade path analyses for Experiment 1: Simple main effects for saccades from initial fixation on the right cheek.**
(PDF)

**S3 Table. Results of pairwise comparisons for the saccade path analyses for Experiment 2a: Main effect of target location.**
(PDF)

**S4 Table. Results of pairwise comparisons for the total fixation duration analyses of Experiment 2b: Main effects of emotion.**
(PDF)

**S5 Table. Results of pairwise comparisons for the total fixation duration analyses of Experiment 2b: Main effects of region of interest.**
(PDF)

**S6 Table. Results of pairwise comparisons for the percentage fixation duration analyses of Experiment 2b: Main effects of emotion.**
(PDF)

**S7 Table. Results of pairwise comparisons for the percentage fixation duration analyses of Experiment 2b: Main effects of region of interest.**
(PDF)

**S1 File. Supplementary results and discussion.**
(PDF)

## Acknowledgments

We are grateful to Hannah Smithson (Oxford University) for devising the calculations for the saccade trajectory analyses, and for help in the initial stages of the experimental design, coding and analyses, to Elaine Stanton (Durham University) for help in processing the eye-tracking data and additional help in coding the experiment, and to Sin-Yee Emma Lip and Holly Naylor (Durham University) for assistance with data collection.

## Author Contributions

**Conceptualization:** Nazire Duran, Anthony P. Atkinson.

**Data curation:** Nazire Duran.

**Formal analysis:** Nazire Duran, Anthony P. Atkinson.

**Investigation:** Nazire Duran.

**Methodology:** Nazire Duran, Anthony P. Atkinson.

**Project administration:** Anthony P. Atkinson.

**Resources:** Anthony P. Atkinson.

**Software:** Nazire Duran, Anthony P. Atkinson.

**Supervision:** Anthony P. Atkinson.

**Visualization:** Nazire Duran, Anthony P. Atkinson.

**Writing – original draft:** Nazire Duran, Anthony P. Atkinson.

**Writing – review & editing:** Nazire Duran, Anthony P. Atkinson.

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
