## [Decision Letter · Decision Letter 0]

17 Mar 2021

Pécs, Hungary

March 15, 2021

PONE-D-21-03786

Foveal processing of emotion-informative facial features

PLOS ONE

Dear Dr. Atkinson,

Thank you for submitting your manuscript to PLOS ONE. After careful consideration, we feel that it has merit but does not fully meet PLOS ONE’s publication criteria as it currently stands. Therefore, we invite you to submit a revised version of the manuscript that addresses the points raised by the Reviewers, listed below.

We look forward to receiving your revised manuscript.

Kind regards,

Joseph Najbauer, Ph.D.

Academic Editor

PLOS ONE

Journal Requirements:

2. Please note that outmoded terms and potentially stigmatizing labels should be changed to more current, acceptable terminology. Examples: “Caucasian” should be changed to “white” or “of [Western] European descent” (as appropriate).

3. We note that Figure 1 and 4 include an image of a participant in the study. 

Reviewers' comments:

Reviewer's Responses to Questions

**Comments to the Author**

1. Is the manuscript technically sound, and do the data support the conclusions?

Reviewer #1: Yes

Reviewer #2: Partly

Reviewer #3: Yes

2. Has the statistical analysis been performed appropriately and rigorously? 

Reviewer #1: Yes

Reviewer #2: Yes

Reviewer #3: Yes

3. Have the authors made all data underlying the findings in their manuscript fully available?

Reviewer #1: Yes

Reviewer #2: Yes

Reviewer #3: Yes

4. Is the manuscript presented in an intelligible fashion and written in standard English?

Reviewer #1: Yes

Reviewer #2: Yes

Reviewer #3: Yes

5. Review Comments to the Author

Reviewer #1: The authors report 3 experiments in which faces with varying emotional expressions are presented (twice under brief presentation, once for longer durations). The authors investigate (1) whether foveating emotion-distinguishing features enhances accuracy of emotion recognition, (2) whether reflexive saccades are targeted at these features, and (3) whether emotion-distinguishing features are looked at longer and whether that relates to better emotion classification performance. Based on these three experiments, the authors conclude that foveal processing of certain features is functional for emotion recognition, but that these features are not automatically sought out.

In general, the paper is well written. The introduction is extensive, previous literature is well described and cited, the results and analysis are elaborately described, and the (sometimes contradictory) findings are well discussed in light of the authors' own experiments and previous work, and the rest of the literature. In conclusion, this paper present a thorough empirical contribution on facial emotion perception. I had no major concerns when reading the paper. I have several minor questions and comments that the authors may wish to address:

1. Can AOI size be presented in degrees as well, and compared to the accuracy obtained with the eye tracker? That is useful information for the reader, to understand whether fixations to each AOI could be well captured.

2. Figures 3 and 8 are difficult to interpret on a small screen (and potentially on a printed A4). While I value showing the full distributions, perhaps the authors could consider a more effective manner to convey the amount of results here (particularly panels b-g).

3. It is unclear to me how total fixation duration is operationalised. Is it the sum of all fixation durations in an AOI? Or is it the sum of all dwell durations from entry to exit of an AOI? I find the usage of fixation duration versus dwell time confusing at times, e.g.:

L. 989: Total dwell time, or dwell time?

L. 1043: Fixation duration, or dwell time?

I suggest using one term and sticking to it (mean total dwell time seems most appropriate).

4. For figure 7 and 8 and the corresponding analyses I suggest reporting in seconds, not milliseconds.

5. Figure 8 could be improved by removing the grey background. Also the units are missing on the x-axes.

Minor things I found:

L. 101-102, l. 640, l.1106-1107: References are in different format from the rest of the paper.

L. 190 "valanced" -> "valenced"?

Reviewer #2: In this paper, the authors describe three experiments that test whether foveating informative regions of emotional faces improves emotion classification accuracy. In Experiments 1a and 2a, they use a brief-fixation paradigm, in which face images are positioned such that a particular feature is shown at the fovea. The results indicated that initial fixation location affected classification performance for specific expressions. In addition, they analyzed observers’ initial saccades, and conclude that informative features are not automatically sought out (and that observers instead may tend to saccade to the feature closest to initial fixation). In Experiment 2b, the authors present the faces for a longer interval and show that fixation durations for specific regions varied depending on the emotional expression, and that fixation durations within some ROIs were associated with accuracy.

The results are interesting, and the authors present a rich dataset and have clearly done a lot of work in analyzing these results. In addition, the discussion of the findings in relation to the previous literature is comprehensive and well-written. However, the presentation of the results could be clearer, as the overall message of the paper gets somewhat lost in the many analyses that they report. In addition, I think further evidence would be needed to support the conclusion that observers do not automatically saccade to informative regions. I have included more detailed comments below:

(1) Experiments 1 and 2a: I understand that many studies using this brief-fixation technique refer to these as “reflexive” saccades, but the latencies seem more consistent with voluntary or memory-guided saccades (on the order of 300-400 ms, and more than 500-600 ms for some observers). It seems unlikely that some of these longer-latency saccades (especially those > 500 ms; potentially after the observer’s button press) would have anything to do with the faces shown. In some ways, it’s not surprising, then, that these saccades do not necessarily target informative regions. I wonder if including longer-latency saccades in the analysis could dilute any effect related to saccades in response to the faces. Would the results change if only the short-latency saccades were analyzed?

(2) In addition, the same question (do observers immediately saccade to informative regions?) could be tested in the longer-duration experiment (2b) by analyzing only the first saccade following face onset, because observers have some opportunity here to gather more information. Does the first saccade target informative regions? If not, this would provide stronger evidence to support the idea that observers’ saccades do not automatically go to informative areas.

(3) Line 284: this sounds as though the next fixation cross appeared immediately after the observer’s button press. Was there any kind of delay (i.e., an inter-trial interval) imposed before the start of the next trial? Is it possible that the saccades analyzed could reflect preparation for the next trial (i.e., a saccade to the next fixation cross)?

(4) In general, the presentation of the results needs to be simplified. One suggestion would be to move results that don’t directly contribute to the conclusions to the supplemental materials (for example, it’s not entirely clear how the exploratory correlations between fixation duration in Exp 2b and accuracy in Exp 2a fit in with the overall conclusions of the paper). Another suggestion for Exp 1 and 2a would be to set up the analysis in a way that more directly tests two competing hypotheses (observers’ initial saccades go to informative regions vs. observers’ saccades go to the nearest feature). Having said that, I appreciate the clear summary of the results in the Discussion section. I have included some specific suggestions for making the presentation of the results clearer:

(a) Figures 2,3,5,6: Since there are many plots/panels, it would be helpful for readers to have a visual reminder of which conditions/comparisons they should be attending to. For example, in Figures 3 and 6, it might be helpful to add arrows to indicating what the hypothesized emotion-distinguishing features are for each expression and/or what the nearest feature(s) are to the initial fixation location.

(b) Figures 3 and 6: adding titles to figure panels would also help (e.g., “Collapsed Across Initial Fixation”, “From the Left Eye”). Figure 8 should have labels as well (which emotional expression is shown in each panel), or perhaps could even be simplied down to a bar graph of correlation values (and putting the scatter plots in the supplemental materials).

(c) Figures 3 and 6 use different color legends for each panel (e.g., cyan represents ‘Left Eye’ in panel (a), but ‘Brow’ in panel (b)), which could be confusing to readers.

(d) Similarly, it’s not clear what information I should be getting out of some of the tables (2,3, 6, 8); the authors could consider moving these to the supplemental materials.

Minor points/typos:

(1) Abstract: The last two experiments are called Experiments 2 and 3 in the abstract, when they’re labeled 2a/2b in the manuscript;

(2) Lines 343-346: It would be helpful to also include a description of what negative values mean here (saccades in the opposite direction to the target?)

(3) The experiment numbers listed in the titles of the supplemental figures don’t match the descriptions underneath

(4) Lines 1106-1107: these references should have numbers to match the rest of the paper

Reviewer #3: I appreciate the opportunity to review this manuscript. This study evaluated whether foveating informative features of briefly presented expressions improves recognition accuracy and whether these features are targeted reflexively when not foveated. The findings suggest that foveal processing of informative features is functional/contributory to emotion recognition, but that they are not automatically sought out when not foveated, and that facial emotion recognition performance is related to idiosyncratic gaze behaviour. The study examines an important topic and is conducted after previous studies in the same area, as a pertinent complementary study. The results of this study have the potential to make a valuable contribution to our understanding of the processing of emotional facial expression perception and their recognition. I have only minor questions and suggestions that could improve the manuscript.

Page 8, there is a very recent study that you probably were not aware of, which could be to my mind interesting to consider in this part as in the discussion: Poncet et al., 2021 (PLoS One. doi: 10.1371/journal.pone.0245777). In this study, the authors studied the link with the pattern of free exploration (no ROIs in statistical analysis) in the correct recognition and the misinterpretation and observed the influence of the task on the participants’ response concerning the emotion classification.

Page 10, in Method/participants section: as for the experiment 2, the experiment 1 has been conducted on young participants which were essentially female studying psychology. First, I did not see the range of the ages, please add this information (which is present in the method part for experiment 2). Second, beyond the potential interrogation on the representativity of such a sample as compared to the global population (but you mention the limitation in the discussion part), I wonder if it could not be suitable to add some information about the literature on the effect of the gender of the observer on the emotional expression processing and/or to include statistical analysis to evaluate the potential influence of it.

In the Design part page 11-12, you indicate to consider the influence of the (initial) “Fixation Location (eyes, brow, cheeks, and mouth)”; later, you indicate that each trial started with a fixation cross located at one of 25 possible locations on the screen that were at 0, 25, or 50 pixels left or right and up or down from the center of the screen, randomly ordered across trials. It is not very clear for me, are these 25 possible locations correspond for each to the fixation locations: eyes, brow, cheeks and mouth respectively? Could you precise it please?

Page 12, you indicate that the keys were labelled A, F and Su and Sa from left to right and the order of these keys remained the same for each participant (A and F keys with the left and the Su and Sa keys with the right hand). If I understand the rational for choosing this configuration to optimize the participants task (no need to look down towards the keyboard during the experiment), I wondered if it could not cause a control problem. Considering that there is a difference in the implication in emotional/facial expression processing at the level of the brain between the right and the left part of the brain, might this choice to use only left hand for some expressions choice and the right for the others and not to have randomized might not cause a bias in the study?

Page 13, line 297, it is indicated “viz.”, is that for “visually”/visualized? Please precise it.

Page 14: “otherwise stated, the data from the left and right sides of the face (when the fixation location was an eye or cheek) were collapsed for the analyses ». Would it be interesting (and does it enable it) to evaluate nevertheless the possible influence of the side (since in the emotional facial expression processing we are side-biased)? You also precise that “participants who had reflexive first saccades on fewer than 20% of the total trials per block (i.e. 20% of 96) in any one of the 4 blocks were removed from further analysis. » Could you indicate how this criterion has been chosen please?

Page 27, Experiment 2, you mention the importance of the information brought by the mouth and the neighbouring wrinkled nose region as informative for the recognition of disgust notably. Would has not been interesting to also consider the nose as an “initial forced fixation location”? (event if you mention later in the manuscript that in partially include a part of the nose area - line 1267-1269 - , and that the nose is considered as the center of the screen in the study and the additional analysis). In addition, you consider its importance since you also use the nose as ROI later.

Page 31, same type of interrogation: I wonder if considering only the central brow is not limited (how did the choice not to include/consider the eyebrows as first forced location has been made?)

Page 34, Saccade path analysis: I am not convinced of the use of the word "effect" (even associated with « small »), with a p>.06. Maybe use “marginal main effect”?.

Page 38, Table 7: For several expressions, the total is not 100% when we add the different percentages (it is frequently less, sometimes almost 1% less), is that due to "approximative rounded" or is that because something else, like a lack or problem of answer in some participants?

Page 47, General Discussion: I think there is a mistake here, line 1087-1089, you mentioned twice the result (different) for hypothesis 2 and not for the 3.

6. PLOS authors have the option to publish the peer review history of their article (what does this mean?). If published, this will include your full peer review and any attached files.

Reviewer #1: No

Reviewer #2: No

Reviewer #3: No

---

## [Author Response · Author response to Decision Letter 0]

17 Aug 2021

Dear Dr Najbauer,

Thank you for considering our manuscript for publication in PLoS One and for giving us the opportunity to respond to the reviewers’ comments. We are grateful to all 3 reviewers for their constructive and helpful feedback, which we respond to in full below and which we have used to revise the paper accordingly. All editorial and reviewer comments are copied below, with our responses in red text underneath each one. Revisions to the main paper are noted below and are highlighted via ‘track changes’ in the document labelled ‘Revised Manuscript with Track Changes'.

Kind regards,

Nazire Duran and Anthony Atkinson 

2. Please note that outmoded terms and potentially stigmatizing labels should be changed to more current, acceptable terminology. Examples: “Caucasian” should be changed to “white” or “of [Western] European descent” (as appropriate). 

On page 10, we now changed ‘Caucasian adults’ to ‘White adults’. 

3. We note that Figure 1 and 4 include an image of a participant in the study.

Images used in figures 1 and 4 are the facial expression images from the Radbound Face Database that were used in the experiment. We did not take pictures of our participants. Permission is not required to reprint these images in scientific publications – see http://www.socsci.ru.nl:8180/RaFD2/RaFD?p=faq

Reviewer #1: The authors report 3 experiments in which faces with varying emotional expressions are presented (twice under brief presentation, once for longer durations). The authors investigate (1) whether foveating emotion-distinguishing features enhances accuracy of emotion recognition, (2) whether reflexive saccades are targeted at these features, and (3) whether emotion-distinguishing features are looked at longer and whether that relates to better emotion classification performance. Based on these three experiments, the authors conclude that foveal processing of certain features is functional for emotion recognition, but that these features are not automatically sought out.

In general, the paper is well written. The introduction is extensive, previous literature is well described and cited, the results and analysis are elaborately described, and the (sometimes contradictory) findings are well discussed in light of the authors' own experiments and previous work, and the rest of the literature. In conclusion, this paper present a thorough empirical contribution on facial emotion perception. I had no major concerns when reading the paper. I have several minor questions and comments that the authors may wish to address:

1. Can AOI size be presented in degrees as well, and compared to the accuracy obtained with the eye tracker? That is useful information for the reader, to understand whether fixations to each AOI could be well captured.

This information is now presented in Table 3 on page 30, along with the following sentence on page 29:

“The average size of each ROI in degrees of visual angle is presented in Table 3. The typical average accuracy of the EyeLink 1000 system is 0.5°, which is much smaller than the sizes of our chosen ROIs, giving us the confidence that any fixation falling within these ROIs will be captured accurately by the eye tracker.”

2. Figures 3 and 8 are difficult to interpret on a small screen (and potentially on a printed A4). While I value showing the full distributions, perhaps the authors could consider a more effective manner to convey the amount of results here (particularly panels b-g).

Figures 3 and 6 have been modified to make the individual panels larger (primarily by now not repeating the x-axis labels and legends for every panel). Figure 8 now no longer contains as many panels/individual plots (the ones taken out have been moved to the Supplementary Materials – Figs. S5 & S6), so the remaining 4 panels/plots are larger and more visible. All figures have been resized to conform to the journal’s guidelines. We have checked the visibility of the individual panels in Figures 3 and 6 and we believe that, although small, they are sufficiently legible both online (at the size they would appear in the published manuscript) and when printed on A4. Additional revisions to Figures 3, 6 & 8 have also been implemented in response to comments from Reviewers 2 and 3 (see below).

3. It is unclear to me how total fixation duration is operationalised. Is it the sum of all fixation durations in an AOI? Or is it the sum of all dwell durations from entry to exit of an AOI? I find the usage of fixation duration versus dwell time confusing at times, e.g.:

L. 989: Total dwell time, or dwell time?

L. 1043: Fixation duration, or dwell time?

I suggest using one term and sticking to it (mean total dwell time seems most appropriate).

Thank you for pointing this out. In the first submitted version of the manuscript we sometimes erroneously used the term ‘dwell time’ instead of ‘total fixation duration’ or similar. Total fixation duration was operationalised as the sum of all fixation durations in an AOI per trial for each participant and emotion, rather than the sum of all dwell durations from entry to exit of an AOI. We have revised our account of this measure in the manuscript, which now reads (pages 28-29):

“For Experiment 2b, we also calculated for each participant the mean total fixation duration for each of 4 regions of interest (ROIs) per emotion, that is, for each participant we summed the durations of all fixations in each ROI per trial, up to the point at which the participant pressed the response button (or, in the very small number of trials where participants responded after face offset, for the full 5 s face duration), and then calculated the average of these total fixation durations for each emotion for each participant.”

In the revised manuscript we have also replaced all erroneous instances of ‘dwell’ or ‘dwell time’ with terms that refer to the actual measure of total fixation duration.

4. For figure 7 and 8 and the corresponding analyses I suggest reporting in seconds, not milliseconds. 

The x-axis values for Figures 7 and 8 are now reported in seconds (also changed in the new Figs. S5 & S6).

5. Figure 8 could be improved by removing the grey background. Also the units are missing on the x-axes. 

The grey background has been removed and the units have been added (seconds – see response to comment 4 above).

Minor things I found:

L. 101-102, l. 640, l.1106-1107: References are in different format from the rest of the paper.

Thank you for pointing this out, these references are now corrected. 

L. 190 "valanced" -> "valenced"?

This has now been corrected.

Reviewer #2: In this paper, the authors describe three experiments that test whether foveating informative regions of emotional faces improves emotion classification accuracy. In Experiments 1a and 2a, they use a brief-fixation paradigm, in which face images are positioned such that a particular feature is shown at the fovea. The results indicated that initial fixation location affected classification performance for specific expressions. In addition, they analyzed observers’ initial saccades, and conclude that informative features are not automatically sought out (and that observers instead may tend to saccade to the feature closest to initial fixation). In Experiment 2b, the authors present the faces for a longer interval and show that fixation durations for specific regions varied depending on the emotional expression, and that fixation durations within some ROIs were associated with accuracy.

The results are interesting, and the authors present a rich dataset and have clearly done a lot of work in analyzing these results. In addition, the discussion of the findings in relation to the previous literature is comprehensive and well-written. However, the presentation of the results could be clearer, as the overall message of the paper gets somewhat lost in the many analyses that they report. In addition, I think further evidence would be needed to support the conclusion that observers do not automatically saccade to informative regions. I have included more detailed comments below:

(1) Experiments 1 and 2a: I understand that many studies using this brief-fixation technique refer to these as “reflexive” saccades, but the latencies seem more consistent with voluntary or memory-guided saccades (on the order of 300-400 ms, and more than 500-600 ms for some observers). It seems unlikely that some of these longer-latency saccades (especially those > 500 ms; potentially after the observer’s button press) would have anything to do with the faces shown. In some ways, it’s not surprising, then, that these saccades do not necessarily target informative regions. I wonder if including longer-latency saccades in the analysis could dilute any effect related to saccades in response to the faces. Would the results change if only the short-latency saccades were analyzed?

We examined first saccades within 1000ms from face offset, following what other investigators (Gamer and colleagues and Kliemann and colleagues) had done in several publications. We hadn’t really considered this really good point raised by the reviewer, though. Following this suggestion, the analysis of these initial saccades was re-run with saccades that happened less than 500ms following face offset for Experiment 2. This analysis produced results that are very similar to the original results and is reported in Supplementary Results and Discussion, on pages 11-13. In the main manuscript, we added:

“This analysis is also repeated with initial saccades that took place within 500ms of face offset, as reported in the Supplementary Results and Discussion. In that analysis, no significant main effects or an interaction were found.” on page 34

and 

“Similar to the analysis of saccades collapsed across initial fixation, the analysis of saccades starting within 500ms of face offset from each initial fixation location is reported in Supplementary Results and Discussions. This analysis yielded similar results to those reported here.” on page 36. 

This suggested analysis was only done on saccades that were made within 500ms after face offset and we did not want to go any lower than this since even with 500ms limit, there was a large drop in the number of valid saccades to be analysed. Going lower than 500ms would limit the statistical power of the analysis. 

(2) In addition, the same question (do observers immediately saccade to informative regions?) could be tested in the longer-duration experiment (2b) by analyzing only the first saccade following face onset, because observers have some opportunity here to gather more information. Does the first saccade target informative regions? If not, this would provide stronger evidence to support the idea that observers’ saccades do not automatically go to informative areas.

Thank you for this excellent suggestion. We have conducted this analysis as suggested and included the results in the Supplementary Results and Discussion. These analyses provide interesting results, in that they are consistent with the hypothesis that first saccades do tend towards emotion-informative facial features when the face is visible for longer (the central brow for anger, the mouth for disgust and surprise). We have added mention of these analyses in the main paper (page 41) as follows:

“Nonetheless, exploratory analyses of the saccade path measures for first saccades after face onset in Experiment 2b, in which the faces were presented for longer, did in fact provide evidence consistent with those initial saccades targeting emotion-informative facial regions (see Supplementary Results and Discussion). Specifically, first saccades were more strongly in the direction of the central brow for angry compared to fearful, surprised and disgusted faces, and more strongly towards the mouth for surprised and disgusted faces than for angry or fearful faces.”

We have also added brief description and discussion of these results in the General Discussion (page 49), as follows:

“In a further analysis, we examined the saccade paths of the first saccades after face onset in the long presentation paradigm of Experiment 2b (reported in the Supplementary Results and Discussion) to more directly assess whether saccades target emotion-informative features. Contrary to the reflexive saccade results from the brief fixation paradigm, these saccades support our hypothesis. More specifically, first saccades after onset were more strongly in the direction of the central brow for angry compared to fearful, surprised and disgusted faces, and more strongly towards the mouth for surprised and disgusted faces than for angry or fearful faces. Therefore, it is possible that medium to low spatial frequency information from the emotion-informative facial features might guide the selection of the next saccade target as long as they are visually available.”

(3) Line 284: this sounds as though the next fixation cross appeared immediately after the observer’s button press. Was there any kind of delay (i.e., an inter-trial interval) imposed before the start of the next trial? Is it possible that the saccades analyzed could reflect preparation for the next trial (i.e., a saccade to the next fixation cross)?

There was no inter-trial interval; the next trial began immediately after the participant’s response. In order to avoid predictive saccades, the fixation location on every trial was made unpredictable by randomly choosing its location on the screen. However, you are right to point out that this might now always preclude the possibility of saccading to a position where the fixation cross is expected. Therefore, we looked at the timescale of events to assess the probability of this. On each trial, the face image is on the screen for 80 ms and the initial saccades that we analysed happened within 1000ms of face offset. The average latency of the saccades analysed here for both Experiment 1 and Experiment 2a are within 400-600ms and the range of the average manual response times (i.e., from face onset of one trial to fixation onset for the next trial) for both Experiments 1 and 2a is 980ms – 1530ms. Therefore, we are confident that the saccades analysed happened well before the next trial began. 

(4) In general, the presentation of the results needs to be simplified. One suggestion would be to move results that don’t directly contribute to the conclusions to the supplemental materials (for example, it’s not entirely clear how the exploratory correlations between fixation duration in Exp 2b and accuracy in Exp 2a fit in with the overall conclusions of the paper).

We agree that the exploratory correlational analyses reported at the end of Experiment 2b do not explicitly address the research questions set out at the beginning, therefore we moved these analyses to the Supplementary Results and Discussion, leaving a note to this effect in the main paper (page 29) as follows:

“In further exploratory analyses, we examined (1) relationships between time spent fixating the mouth or central brow of the different emotional faces in Experiment 2b and recognition accuracy with enforced brief fixation on those features in Experiment 2a, given that the same participants took part in both experiments, and (2) whether fixation duration in any of the 4 ROIs, regardless of the emotion expressed on the face, was related to accuracy in classifying any of the emotions. These analyses are reported in the Supplementary Results and Discussion.”

Another suggestion for Exp 1 and 2a would be to set up the analysis in a way that more directly tests two competing hypotheses (observers’ initial saccades go to informative regions vs. observers’ saccades go to the nearest feature. Having said that, I appreciate the clear summary of the results in the Discussion section. I have included some specific suggestions for making the presentation of the results clearer:

Thank you very much for this suggestion and we do agree that simplifying the results would be beneficial for the reader. Regarding the suggestion for Experiment 1 and 2a, our initial hypothesis regarding the target of initial saccades was that they will target informative features, based on previous literature. However, following analysis of the results we obtained, we noticed that the saccades might be more likely to target nearby features. In other words, targeting nearby features was a conclusion we reached following analysis so we feel that it might not be best practice to run such analysis following data collection since it might bias the analysis post-hoc. 

(a) Figures 2,3,5,6: Since there are many plots/panels, it would be helpful for readers to have a visual reminder of which conditions/comparisons they should be attending to. For example, in Figures 3 and 6, it might be helpful to add arrows to indicating what the hypothesized emotion-distinguishing features are for each expression and/or what the nearest feature(s) are to the initial fixation location. 

Figures 3 and 6 have been updated to include arrows indicating the most emotion-informative (‘diagnostic’) facial features for each emotion, i.e., those features that related to specific hypotheses tested in the reported analyses.

(b) Figures 3 and 6: adding titles to figure panels would also help (e.g., “Collapsed Across Initial Fixation”, “From the Left Eye”). Figure 8 should have labels as well (which emotional expression is shown in each panel), or perhaps could even be simplied down to a bar graph of correlation values (and putting the scatter plots in the supplemental materials). 

Figures 3 and 6 have been updated with titles for each panel, as suggested. Figure 8 has now been updated to show only the scatter plots of the correlations between fixation duration on the mouth and emotion recognition accuracy in Experiment 2b, and titles have been added for each scatterplot, as suggested. Scatterplots for all other correlations have been moved to the Supplementary Materials (Fig S5 and Fig S6).

(c) Figures 3 and 6 use different color legends for each panel (e.g., cyan represents ‘Left Eye’ in panel (a), but ‘Brow’ in panel (b)), which could be confusing to readers. 

The colour legends have been updated in Figures 3 and 6 to be consistent across all panels. 

(d) Similarly, it’s not clear what information I should be getting out of some of the tables (2,3, 6, 8); the authors could consider moving these to the supplemental materials. 

Thank you for this suggestion, these tables have been moved into the Supplementary Materials and references to them in text are corrected.

Minor points/typos:

(1) Abstract: The last two experiments are called Experiments 2 and 3 in the abstract, when they’re labeled 2a/2b in the manuscript; - We have corrected this.

(2) Lines 343-346: It would be helpful to also include a description of what negative values mean here (saccades in the opposite direction to the target?).

We have added the following on page 15 of the revised manuscript:

“Negative saccade path values, on the other hand, indicate a saccade that is going in the opposite direction of the possible saccade path.”

(3) The experiment numbers listed in the titles of the supplemental figures don’t match the descriptions underneath – 

We have corrected the experiment numbers in the supplemental figures. 

(4) Lines 1106-1107: these references should have numbers to match the rest of the paper 

All references are updated to match the required style. 

Reviewer #3: I appreciate the opportunity to review this manuscript. This study evaluated whether foveating informative features of briefly presented expressions improves recognition accuracy and whether these features are targeted reflexively when not foveated. The findings suggest that foveal processing of informative features is functional/contributory to emotion recognition, but that they are not automatically sought out when not foveated, and that facial emotion recognition performance is related to idiosyncratic gaze behaviour. The study examines an important topic and is conducted after previous studies in the same area, as a pertinent complementary study. The results of this study have the potential to make a valuable contribution to our understanding of the processing of emotional facial expression perception and their recognition. I have only minor questions and suggestions that could improve the manuscript.

Page 8, there is a very recent study that you probably were not aware of, which could be to my mind interesting to consider in this part as in the discussion: Poncet et al., 2021 (PLoS One. doi: 10.1371/journal.pone.0245777). In this study, the authors studied the link with the pattern of free exploration (no ROIs in statistical analysis) in the correct recognition and the misinterpretation and observed the influence of the task on the participants’ response concerning the emotion classification.

Thanks for bringing this interesting study to our attention.

The following sentence has been added to page 8 of the revised manuscript, to reflect the findings of recent research:

“…except for one recent study [41], which found that when the observers misclassified expressions, they explored regions of the face that supported the recognition of the mistaken expression” 

Additionally, to compare our relevant findings to the findings of Poncet et al., this was added to page 47:

“On the other hand, in contrast to our findings, Poncet et al. [41] found the mouth region to be associated with increased misclassification of anger. One potential reason for the contradictory results might be difference in the AUs used by the models in the two studies. While the AUs used by the models in the present experiments for angry expressions included the lip tightener and the lip presser, models in [41] used a variety of AUs leading to an open mouth for angry expressions. Since the open mouth is a shared feature in angry and disgusted expression, this might have led to confusions between these expressions while in our study the most distinctive feature between anger and disgust was the open mouth for disgust and the pressed lips for anger.”

Page 10, in Method/participants section: as for the experiment 2, the experiment 1 has been conducted on young participants which were essentially female studying psychology. First, I did not see the range of the ages, please add this information (which is present in the method part for experiment 2) –Second, beyond the potential interrogation on the representativity of such a sample as compared to the global population (but you mention the limitation in the discussion part), I wonder if it could not be suitable to add some information about the literature on the effect of the gender of the observer on the emotional expression processing and/or to include statistical analysis to evaluate the potential influence of it 

Thank you for pointing this out and we agree about the issue about the representativeness of a population made up of entirely psychology undergraduates of a narrow age range. (Age ranges are now added to all experiments.) Regarding the statistical analysis of potential gender differences, we don’t believe that a meaningful statistical analysis can be conducted on the unbalanced sample that we have (in all cases, less than a quarter of the sample are males). However, we do agree that this should be mentioned as a limitation and therefore the following has been added to the discussion section of the manuscript on pages 54-55:

“Certain limitations of the experiments imposed by the participant demographics need to be acknowledged. The majority of the participants in both studies were female. Previous research suggested a female superiority in emotion recognition [77–81] which was sometimes only reflected in faster RTs rather than accuracy [82,83]. Additionally, previous research suggests that females spend longer looking at the eye regions compared to males [78]. This female superiority is suggested to be most prominent for negative emotions [80,82]. However, while the literature on gender differences in emotion recognition is vast, the results are not always unequivocal. Some recent studies found no difference in the emotion recognition performance or eye movement patterns between young males and females [41,77,83] whose ages were close to the age of our sample. These groups of researchers suggest that that female superiority in emotion recognition is more prominent in older participants with older females better at recognizing emotions compared to older men and spending more time looking at the eye region more compared to older men. However, despite the equivocal results in the literature regarding gender differences in emotion recognition performance, it must be noted that the predominantly female sample in the experiments reported here might mean that the emotion recognition accuracy was higher than what can be expected in a more diverse population. Additionally, inclusion of more male participants might decrease the suggested informativeness of the eye region since males are suggested to look less at the eye region of faces.”

In the Design part page 11-12, you indicate to consider the influence of the (initial) “Fixation Location (eyes, brow, cheeks, and mouth)”; later, you indicate that each trial started with a fixation cross located at one of 25 possible locations on the screen that were at 0, 25, or 50 pixels left or right and up or down from the center of the screen, randomly ordered across trials. It is not very clear for me, are these 25 possible locations correspond for each to the fixation locations: eyes, brow, cheeks and mouth respectively? Could you precise it please? – 

We admit that the description in the Methods could have been clearer in this regard, and we have revised that description accordingly (see below). In all experiments, the face was presented on the screen such that one particular location or feature of the face image (namely, one of the eyes, the central brow, the left or right cheek or the centre of the mouth) was aligned with the participant’s fixation location on the screen. That initial fixation location on the screen (i.e., the location of the fixation cross that appeared at the start of each trial and onto which the participant had to fixate in order to trigger the onset of the face) was determined randomly on a trial-by-trial basis from one of 25 possible screen locations (0, 25, or 50 pixels left or right and up or down from the centre of the screen). This random selection of the initial fixation location was done so that the location of the face and thus of the individual features within that face could be made unpredictable and so that the participants would not predict which facial feature would be presented at fixation. If the face had always been presented in the same central location on every trial, the participants would have been able to guess that a fixation cross on the top right will refer to the right eye and a fixation cross in the lower visual field will correspond to a mouth.

We have revised the text on page 12 of the revised manuscript to further clarify the description:

“Each trial started with a fixation cross located at one of 25 possible locations on the screen. This was to make both the exact screen location of the fixation cross and the to-be-fixated facial feature unpredictable. These 25 possible locations for the fixation cross were at 0, 25, or 50 pixels left or right and up or down from the center of the screen. These fixation-cross positions were randomly ordered across trials. Following the presentation of the fixation cross on one of these randomly selected screen locations, a face image was presented so that one of the facial features of interest was aligned with the location of that fixation cross. Faces were presented in a gaze-contingent manner on each trial: The participants needed to fixate within 30 pixels (1.16 degrees of visual angle) of the fixation cross for 6 consecutive eye-tracking samples following which a face showing one of the four target expressions was presented. The facial features of interest (which henceforth we refer to as ‘fixation locations’) were the left or right eye, the central brow, the centre of the mouth, and locations on the left and right cheeks. . These initial fixation locations can be seen in Fig 1.”

Page 12, you indicate that the keys were labelled A, F and Su and Sa from left to right and the order of these keys remained the same for each participant (A and F keys with the left and the Su and Sa keys with the right hand). If I understand the rational for choosing this configuration to optimize the participants task (no need to look down towards the keyboard during the experiment), I wondered if it could not cause a control problem. Considering that there is a difference in the implication in emotional/facial expression processing at the level of the brain between the right and the left part of the brain, might this choice to use only left hand for some expressions choice and the right for the others and not to have randomized might not cause a bias in the study?

The two prominent hypotheses regarding the lateralization of emotions state either that the right hemisphere is involved in recognition of all emotions (eg. Burt & Perrett, 1997) or the right hemisphere is involved in the recognition of negatively valanced emotions (eg. Jansari, Tranel & Adolphs, 2000). According to the hypothesis that right hemisphere is specialised in processing of all emotions, we would expect that the emotions that need to be responded to with the left hand to be recognised better. In all experiments reported here, angry and fearful responses were associated with the left hand; however, fear is always the emotion that is recognised worst. Also, it is possible that production of emotion labels to produce a response to the visually presented emotions might rely more on the left hemisphere (Broca’s area). Additionally, among the expressions/emotions investigated in the reported experiments, anger, fear, disgust and sadness are all negatively valanced emotions and according to the valance hypothesis, should be recognized by the right hemisphere. While the low recognition rates for sad and disgusted expressions can be attributed to the right hemisphere dominance for negatively valanced emotions, the recognition of fearful expressions, which is also negatively valanced, cannot be since the response associated with fear was completed with the left hand. Additionally, the pattern of recognition accuracy we report in these experiments are in line with what has already been found in the literature. Thus, for all these reasons we are confident that not randomizing the response buttons is very unlikely to have produced a bias in emotion recognition responses.

Page 13, line 297, it is indicated “viz.”, is that for “visually”/visualized? Please precise it. 

‘viz.’ is a not uncommonly used abbreviation of a Latin term, used in English to mean ‘namely’ or ‘in other words’. We have replaced ‘viz’ with the more commonly used ‘i.e.’, now on page 14. 

Page 14: “otherwise stated, the data from the left and right sides of the face (when the fixation location was an eye or cheek) were collapsed for the analyses ». Would it be interesting (and does it enable it) to evaluate nevertheless the possible influence of the side (since in the emotional facial expression processing we are side-biased)? 

Thank you for this comment and we acknowledge the prevalent finding in the literature that there is a perceptual bias to the left side of the face for many face-related tasks including expression recognition whereby performance of the observers is influenced more by the left side of the face (Burt & Perrett, 1997; Butler et al., 2005; Innes, Burt, Birch & Hausmann, 2015). Due to this functional difference between the left and right sides of the face, the analyses reported here, which collapsed across initial fixation location, such as the analysis of initial fixation location on emotion recognition accuracy, might be masking the potential greater informativeness of the left side of the face (left eye and left cheek) over the right side of the face. Additionally, the analysis of the total fixation duration which treats the left eye and the right eye as a unitary ROI might be masking a leftward bias in eye movements. However, we also would like to point out the following. In our previous work with the brief-fixation paradigm (Atkinson & Smithson, 2020) we have already reported that reflexive first saccades tended towards the left and centre of the face rather than preferentially targeting emotion-distinguishing features. Although we agree that additional analyses in the present paper to examine differences between the left and right sides of the face would be interesting, such analyses would not address the research questions or hypotheses we wanted to investigate in this present series of experiments and would add more analysis to the already crowded results section.

You also precise that “participants who had reflexive first saccades on fewer than 20% of the total trials per block (i.e. 20% of 96) in any one of the 4 blocks were removed from further analysis. » Could you indicate how this criterion has been chosen please?

The minimum of 20% of the total 96 trials per block having ‘valid’ first saccades was chosen to strike a balance between having enough trials per condition and not wanting to exclude the data for too many participants. Setting the threshold higher would have included more participants but with them having very few or no trials for at least 1 condition; setting the threshold lower would have reduced the number of participants and thus the power of the experiments. We have added the following statement in the main paper (page 14, following on from the cited sentence):

“This threshold was chosen to strike a balance between having enough trials per condition and not wanting to exclude the data for too many participants.”

Page 27, Experiment 2, you mention the importance of the information brought by the mouth and the neighbouring wrinkled nose region as informative for the recognition of disgust notably. Would has not been interesting to also consider the nose as an “initial forced fixation location”? (event if you mention later in the manuscript that in partially include a part of the nose area - line 1267-1269 - , and that the nose is considered as the center of the screen in the study and the additional analysis). In addition, you consider its importance since you also use the nose as ROI later.

Thank you for this comment and we agree that considering the nose as an initial fixation location would have been interesting. However, there were several considerations for us while choosing the initial fixation locations in this study: (1) The initial fixation locations are chosen to be equidistant and symmetrical. This was done so that at a fixation on, for example, the brow, the spatial-frequency (SF) filtering of the mouth would be equal to the SF filtering at the brow when the mouth is the initial fixation location. The nose is a big facial feature that spans a larger area of the face compared to other features and the nose can be divided into further regions as done by, for example, Schurgin et al. (2014). According to their findings, not all regions of the nose are equally fixated upon, with the most fixations falling on the lower nose region. If we were to consider this section of the nose as an initial fixation location, the distance from this region to the mouth would be shorter compared to the distance between the lower nose and the eyes, This in turn would have created an issue while comparing the accuracy at the nose to the mouth and the eyes since SF filtering in these two comparison conditions would have been unequal. (2) Additionally, the initial fixation locations were chosen to be as specific and as close to the informative regions identified by Smith et al. (2005). For the expressions used in this study, the most distinctive informative regions where high spatial frequency visual information was present (and therefore foveal processing would be more beneficial) were the eyes (fear & surprise), brow (anger) and the mouth (disgust & surprise). While the nose is informative for disgust, the main action unit (AU) that is informative for disgust, especially when this expression needs to be discriminated from anger, seems to be the upper lip raiser (Jack et al., 2009). From the mouth initial fixation, our participants could have accessed visual information from the nose, therefore we believed it important to discuss the possible contribution of visual information from the nose. Additionally, choosing a single region of the nose as the initial fixation location that would be considered informative would be difficult since different regions of the nose can be informative for different expressions, for example, Poncet et al. (2021) found that while the top of the nose is more informative for anger, the mid-nose region is more informative for disgust. 

Regarding the partial inclusion of the nose region (now on on page 51 of the revised manuscript), the central brow region selected in the study sits above the nasion and can be considered more a part of the forehead than the nose itself. To ease this confusion, we have revised the relevant sentence by changing ‘overlapping’ to ‘partially overlapping’.

With respect to the regions of interest for the fixation duration analyses of Experiment 2b, the ROI choice was guided by what has been used in previous studies, to allow for effective comparison between our study and the literature. The nose region is a common ROI in those previous studies. Additionally, we considered the nose ROI as a marker/proxy to test the center-of-gravity effect. The nose is a frequently fixated region of the face particularly for the first few fixations. 

Page 31, same type of interrogation: I wonder if considering only the central brow is not limited (how did the choice not to include/consider the eyebrows as first forced location has been made?)

The reason for this is partially explained in the comment above. Additionally, even though we did not consider the eyebrows, the action of the brow lowerer causes the most distinctive change in the central brow. Another reason that makes considering the eyebrows as initial fixation locations difficult is due to the variation of the position of the eyebrows across emotions. For example, for surprised and fearful expressions the eyebrows are likely to be raised while for anger and disgust the eyebrows are likely to be lowered. Therefore, using a single fixation location at the left or right eyebrow would mean that on some expressions, participants will fixate the eyebrow and on other expressions they will fixate neighbouring skin. Another reason for wanting to focus on the central brow region instead of the whole eyebrow area is that in most previous studies, the lower central brow region has been considered as part of the larger ‘eyes ROIs’ which doesn’t allow for the investigation of the more specific contribution of this area, which is profoundly affected by the action of the brow lowerer. This can further be seen in Figure 2B and C of Poncet et al. (2021), which shows that it is largely the central brow region that received more fixations compared to other expressions. 

Page 34, Saccade path analysis: I am not convinced of the use of the word "effect" (even associated with « small »), with a p>.06. Maybe use “marginal main effect”?.

This suggested change has been implemented. 

Page 38, Table 7: For several expressions, the total is not 100% when we add the different percentages (it is frequently less, sometimes almost 1% less), is that due to "approximative rounded" or is that because something else, like a lack or problem of answer in some participants?

Thank you for pointing this out. We checked the values and the error was caused by 40 missing trials. The table (now Table 5 following revisions) is now revised to take into account these missing trials and values for all expressions now total to 100%. 

Page 47, General Discussion: I think there is a mistake here, line 1087-1089, you mentioned twice the result (different) for hypothesis 2 and not for the 3.

Thanks for pointing out this error. This has been corrected, as follows (page 43 of the revised manuscript):

“We found circumscribed support for hypotheses 1, 3 and 4, and only very limited support for hypothesis 2.”

---

## [Decision Letter · Decision Letter 1]

18 Nov 2021

Foveal processing of emotion-informative facial features

PONE-D-21-03786R1

Dear Dr. Atkinson,

We’re pleased to inform you that your manuscript has been judged scientifically suitable for publication and will be formally accepted for publication once it meets all outstanding technical requirements.

Kind regards,

José A Hinojosa, Ph.D.

Academic Editor

PLOS ONE

Additional Editor Comments (optional):

Reviewers' comments:

Reviewer's Responses to Questions

**Comments to the Author**

1. If the authors have adequately addressed your comments raised in a previous round of review and you feel that this manuscript is now acceptable for publication, you may indicate that here to bypass the “Comments to the Author” section, enter your conflict of interest statement in the “Confidential to Editor” section, and submit your "Accept" recommendation.

Reviewer #1: All comments have been addressed

Reviewer #2: All comments have been addressed

Reviewer #3: All comments have been addressed

2. Is the manuscript technically sound, and do the data support the conclusions?

Reviewer #1: (No Response)

Reviewer #2: (No Response)

Reviewer #3: Yes

3. Has the statistical analysis been performed appropriately and rigorously? 

Reviewer #1: (No Response)

Reviewer #2: (No Response)

Reviewer #3: Yes

4. Have the authors made all data underlying the findings in their manuscript fully available?

Reviewer #1: (No Response)

Reviewer #2: (No Response)

Reviewer #3: Yes

5. Is the manuscript presented in an intelligible fashion and written in standard English?

Reviewer #1: (No Response)

Reviewer #2: (No Response)

Reviewer #3: Yes

6. Review Comments to the Author

Reviewer #1: (No Response)

Reviewer #2: (No Response)

Reviewer #3: The authors have adressed all my comments. The paper has been significantly improved after revising.

I juste noticed a typo line 1358, page 55 (in the revised version without marks):

"These groups of researchers suggest that that female superiority in [...]" (there are two "that").

I really thank the authors for this beautiful work.

7. PLOS authors have the option to publish the peer review history of their article (what does this mean?). If published, this will include your full peer review and any attached files.

Reviewer #1: No

Reviewer #2: No

Reviewer #3: No

---

## [Editor Report · Acceptance letter]

23 Nov 2021

PONE-D-21-03786R1 

Foveal processing of emotion-informative facial features 

Dear Dr. Atkinson:

I'm pleased to inform you that your manuscript has been deemed suitable for publication in PLOS ONE. Congratulations! Your manuscript is now with our production department. 

Kind regards, 

on behalf of

Dr. José A Hinojosa 

Academic Editor

PLOS ONE